# Cryo-EM structure of human eIF5A-DHS complex reveals the molecular basis of hypusination-associated neurodegenerative disorders

Elżbieta Wątor [1,2], Piotr Wilk [1], Artur Biela [1], Michał Rawski [1], Krzysztof M. Zak[1], Wieland Steinchen[3], Gert Bange [3,4], Sebastian Glatt [1] & Przemysław Grudnik [1] ✉

Hypusination is a unique post-translational modification of the eukaryotic translation factor 5A (eIF5A) that is essential for overcoming ribosome stalling at polyproline sequence stretches. The initial step of hypusination, the formation of deoxyhypusine, is catalyzed by deoxyhypusine synthase (DHS), however, the molecular details of the DHS-mediated reaction remained elusive. Recently, patient-derived variants of DHS and eIF5A have been linked to rare neurodevelopmental disorders. Here, we present the cryo-EM structure of the human eIF5A-DHS complex at 2.8 Å resolution and a crystal structure of DHS trapped in the key reaction transition state. Furthermore, we show that disease-associated DHS variants influence the complex formation and hypusination efficiency. Hence, our work dissects the molecular details of the deoxyhypusine synthesis reaction and reveals how clinically-relevant mutations affect this crucial cellular process.

The eukaryotic translation factor 5A (eIF5A) plays a pivotal role during translation[1,2]. It is the only cellular protein known to undergo hypusination, a unique post-translational modification of a conserved lysine residue (Lys50 in human eIF5A). Hypusination is essential to resolve ribosomal stalling during the translation of proline-rich polypeptides[3,4]. Recent findings show that the hypusination of eIF5A plays a role in key cellular processes, including autophagy[5], senescence, polyamine homeostasis, and the determination of helper T cell lineages[6]. Significantly, misregulation of hypusination has been linked to pathological conditions, including cancer and diabetes. The hypusination pathway, therefore, represents an attractive molecular target for therapeutic interventions[7–14].

Hypusination involves two distinct enzymatic steps[14]. During the first, rate-limiting, step, deoxyhypusine synthase (DHS) catalyzes the NAD-dependent transfer of the 4-aminobutyl moiety from spermidine (SPD) to the lysine side chain, forming deoxyhypusine[15,16]. Subsequently, deoxyhypusine is hydroxylated to hypusine (N6-(4-amino-2-hydroxybutyl)lysine) by deoxyhypusine hydroxylase (DOHH)[17]. Structural studies have provided some insights into the hypusination reaction. X-ray crystal structures of the human DHS apoenzyme and structures of the inhibitor (GC7)- and spermidine-bound DHS[13,16,18,19] show the architecture of the protein's composite active site and resolve SPD binding and the molecular basis of inhibition. Furthermore, the X-ray and cryo-EM structures of eIF5A bound to the translating ribosome have revealed how hypusine contributes to the rescue of stalled ribosomes. By stabilizing the productive position of P-tRNA, hypusinated eIF5A facilitates the transfer of the nascent chain from P-tRNA to A-tRNA[20,21]. However, no structural information on substrate-bound complexes during the hypusination reaction, i.e., eIF5A-DHS and eIF5A-DOHH, is yet available.

[1]Małopolska Centre of Biotechnology, Jagiellonian University, Kraków, Poland. [2]Doctoral School of Exact and Natural Sciences, Jagiellonian University, Kraków, Poland. [3]Philipps-University Marburg, Center for Synthetic Microbiology (SYNMIKRO) & Faculty of Chemistry, Marburg, Germany. [4]Max Planck Institute for Terrestrial Microbiology, Molecular Physiology of Microbes, Marburg, Germany. ✉e-mail: przemyslaw.grudnik@uj.edu.pl

Ganapathi et al. recently identified disease-causing mutations in the human DHS encoding gene (*dhps*) and described the cellular consequences that result from these mutations[22]. The authors characterized a genetic syndrome, namely DHS deficiency, and report five affected individuals from four independent families, who carry biallelic variants in *dhps*. The clinical features common to most patients are developmental delay and intellectual disability, abnormalities in muscle tone, clinical seizures, decreased coordination, walking and locomotive and behavioral difficulties.

Herein, we uncover the mechanism of the first rate-limiting step of the hypusination reaction catalyzed by DHS using a hybrid structural biology approach. We sought to elucidate the impact of pathological mutations on the structure and function of proteins involved in the deoxyhypusination complex. We present the single-particle cryo-EM structure of the human eIF5A-DHS complex, which provides mechanistic insights into the deoxyhypusination reaction. Furthermore, we used ambient-temperature X-ray crystallography to solve the structure of DHS trapped in the reaction transition state, in which the postulated form of deoxyhypusinated DHS is directly visible. Moreover, we complement our structural work with thorough biochemical analyses of clinically-relevant DHS variants to reveal the molecular impacts of these mutations. Our data show that clinical mutations associated with neurodegeneration impair the first step of hypusination by affecting the structure and activity of DHS, spermidine binding, the eIF5A-DHS interaction, and the overall efficiency of hypusination.

## Results

### Cryo-EM structure of the eIF5A-DHS$^{K329A}$ complex

To understand how the binding of eIF5A and DHS facilitates the deoxyhypusination reaction, we used single-particle cryo-EM to determine the structure of the human eIF5A-1-DHS complex at an overall resolution of 2.8 Å (Fig. 1a, Supplementary Fig. 1 and Supplementary Table 1). We reconstituted and trapped the complex using individually purified components, namely eIF5A-1 and the catalytically dead variant DHS$^{K329A}$ (we refer to the complex as eIF5A-DHS$^{K329A}$). The alanine substitution of the catalytic lysine rendered an enzymatically inactive DHS yet still capable of binding the cofactor NAD and both substrates, namely SPD and eIF5A.

The complex structure is composed of a single eIF5A monomer bound to a DHS tetramer (Fig. 1b). We observed well-resolved density in the core of DHS, but in some peripheral regions, the map quality appears weaker (e.g., surface regions and the C-terminal OB-fold domain of eIF5A) (Supplementary Fig. 1). The quality of the reconstruction allowed us to unambiguously place most amino acids of both complex components. Similarly, to previously determined DHS crystal structures[16,18,23], individual protomers differ in the length of the defined density for their N-termini and loop regions (aa 79–83), which were often poorly ordered. Remarkably, however, the loop of chain A is significantly better structured in the cryo-EM structure of eIF5A-DHS$^{K329A}$, compared to the crystal structure of DHS (Supplementary Fig. 2a). This difference may be a direct consequence of the adjacent binding of eIF5A and suggests that eIF5A binding may stabilize this chain. The entire N-terminal domain of eIF5A is well-defined in the Coulomb potential map with a well-resolved modifiable lysine residue (Lys50) (Fig. 1c). The C-terminal domain of eIF5A (aa 87-150) is not stabilized by any interaction with DHS and thus remains rather poorly defined (Supplementary Fig. 2b). The overall structure suggests that eIF5A-DHS complex formation is favored by charge complementarity (Fig. 1d).

### eIF5A binding to DHS unmasks the active site

The cryo-EM structure reveals that the DHS tetramer does not undergo pronounced structural rearrangements upon the binding of eIF5A. The overall architecture of DHS in the complex with eIF5A is very similar to that observed in crystal structures of human DHS[16,18,24]. Likewise, the DHS dimer (chains C and D) opposite the eIF5A binding site superposes almost perfectly on the unliganded DHS (RMSD 0.86 Å). Of note, we did observe a 5° rotation of the DHS half that interacts with eIF5A (chains A and B) (Supplementary Fig. 2c). Furthermore, we measured a 7° difference in the relative orientation of the DHS outermost helix (chain B) that flanks the binding site of eIF5A.

The structure also confirms that only one eIF5A monomer interacts with the DHS tetramer as proposed in previous biochemical studies[2]. The binding interface is located between two DHS protomers, which almost equally contribute to eIF5A binding (interaction surface with chain A 489.6 Å$^2$ and chain B 564.2 Å$^2$) and can be explained by DHS active site complementation[25]. Overall the eIF5A interface constitutes a large surface area of ~1054 Å$^2$, which covers approximately 12% of the total surface of eIF5A[25]. This interaction is mediated mainly by a 3-stranded β-sheet formed by the N-terminal domain of eIF5A (Fig. 2a, b). The complex is stabilized by an extended network of specific hydrogen bonds with a significant contribution of less specific hydrophobic contacts (Fig. 2c). Hence, our structural data are consistent with previous findings showing that almost the entire N-terminal domain of eIF5A is needed for effective deoxyhypusination[3]. Our data thus suggest that this requirement reflects the contribution of this domain to the interaction with DHS.

Furthermore, the structure explains how eIF5A binding unmasks the DHS active site. In the absence of eIF5A, the so-called *ball* from the *ball-and-chain* motif in the N-terminal helix of DHS (residues 11–17) covers the entrance to the active site of the enzyme, maintaining a negative surface potential[6,24] (Fig. 1d). The highly conserved hypusination loop of eIF5A ($^{46}$SKTG**K**HGHA[26]) enters the DHS active site through a distinct tunnel and is stabilized mainly by three helices at the DHS dimer interface (Fig. 2). The hypusination loop thereby unlocks the entrance to the active site by displacing the *ball* motif. Of note, the *ball* motif is still clearly present in its previously described position in the remaining three unoccupied active sites of DHS (Supplementary Fig. 2d).

Finally, we observe that NAD binds to all four DHS active sites, while the SPD moiety binds to three—all but the one occupied by eIF5A. Indeed, SPD and Lys50 of eIF5A occupy a similar position within the active site, potentially explaining their exclusive binding. Of note, the density for the SPD ligand is significantly less defined than the density for NAD (Supplementary Fig. 2e).

### HDX-MS reveals alterations in the DHS tetramer upon eIF5A binding

As an alternate approach to understanding the interactions between eIF5A and DHS in solution, we used **h**ydrogen-**d**euterium e**x**change coupled with **m**ass **s**pectrometry (HDX-MS). This method detects conformational changes that occur upon complex formation[27]. We proceeded to compare the HDX profiles of individual eIF5A and DHS proteins with that of the eIF5A-DHS complex (Fig. 2d).

For eIF5A, reduced HDX was apparent for residues 32–77 encompassing most of its N-terminal domain and a small portion (residues 97–101) in the C-terminal domain (Supplementary Fig. 3). The strong reduction in HDX of the former is in good agreement with the cryo-EM structure of the complex, as it contains the hypusination loop protruding into the DHS tetramer (Fig. 2d). Increased HDX mainly clustered in the C-terminal domain (residues 109–114, 120–122, 124–126, 144–151) and a short helical turn in the N-terminal domain (residues 24-30, Supplementary Figs. 3b, c). For some peptides located in these areas, we observed a bimodal distribution of the ion isotope clusters during HDX (Supplementary Figs. 5a–c & 6) indicating the presence of different stable conformations or/and transition between them[28]. For two peptides from the N-terminal domain of eIF5A (residues 32-42 and 41-60), the fast-exchanging species was dominant after 10,000 s of HDX for individual eIF5A, but not for eIF5A in complex with DHS$^{K329A}$ suggesting a restriction of conformational flexibility by DHS

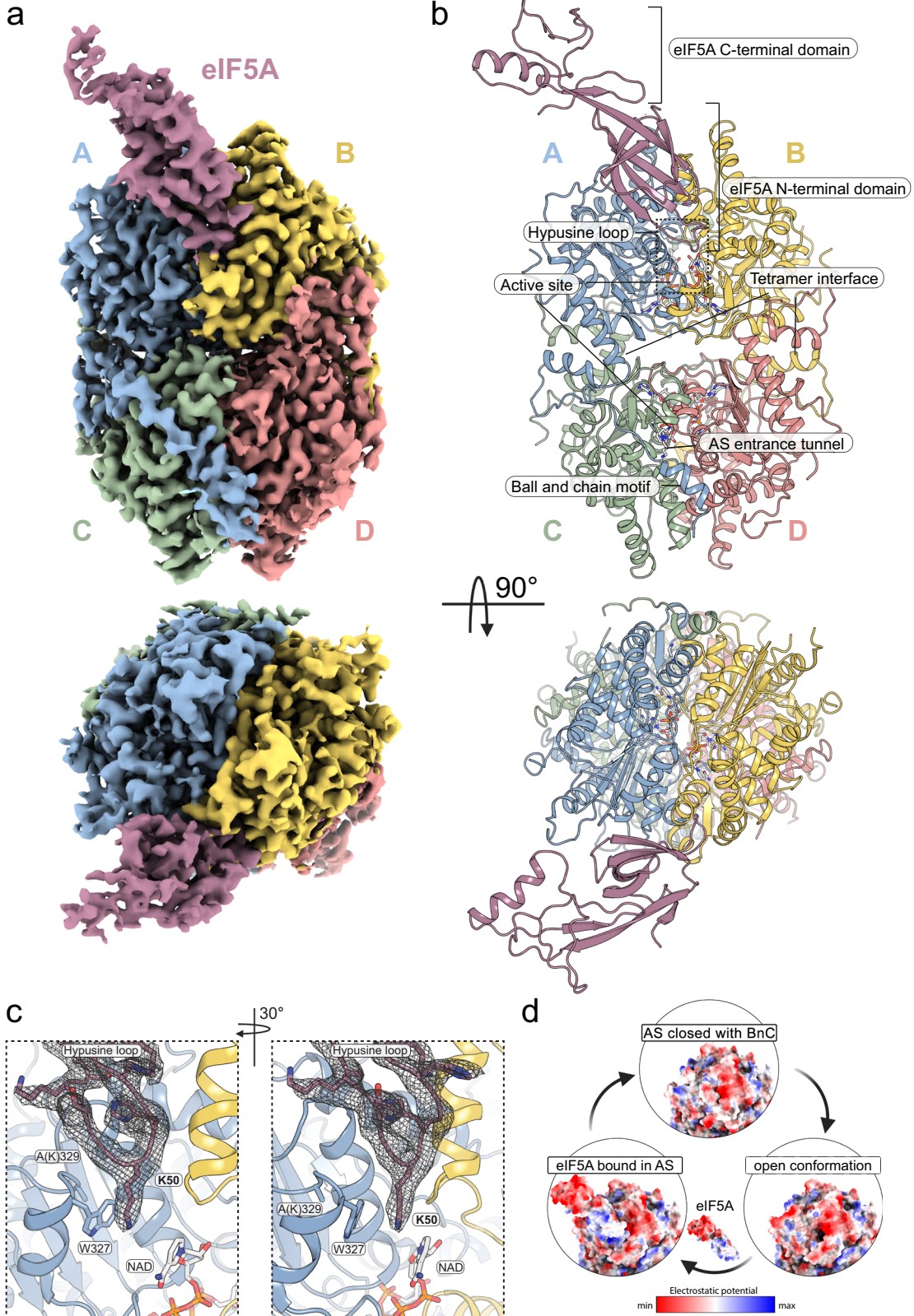

**Fig. 1 | Overview of the eIF5A-DHS complex. a** Cryo-EM map and **b** corresponding molecular model of eIF5A-DHS$^{K329A}$ complex in cartoon representation shown from two perpendicular perspectives (upper *vs.* lower panels). **c** Close-up view of the eIF5A binding site. The cryo-EM map (black mesh at 6σ) is countered around the hypusination loop (dark pink) placed between monomers A (blue) and B (yellow) of the DHS. Modifiable lysine (K50) and rotatable W327 are shown as sticks. Two views rotated by 30 degrees are shown for clarity. NAD is shown in the redox centre.

**d** Surfaces of eIF5A-DHS complex components are colored by electrostatic potential. The DHS in an open conformation exposes a negatively charged (red) patch that is complementary to the positively charged (blue) N-terminal domain of eIF5A, which facilitates the binding of eIF5A to the active site (AS). Upon dissociation of eIF5A, the negative patch at the entrance to the DHS active site can be masked by a negatively charged DHS N-terminal *ball-and-chain* (BnC) motif.

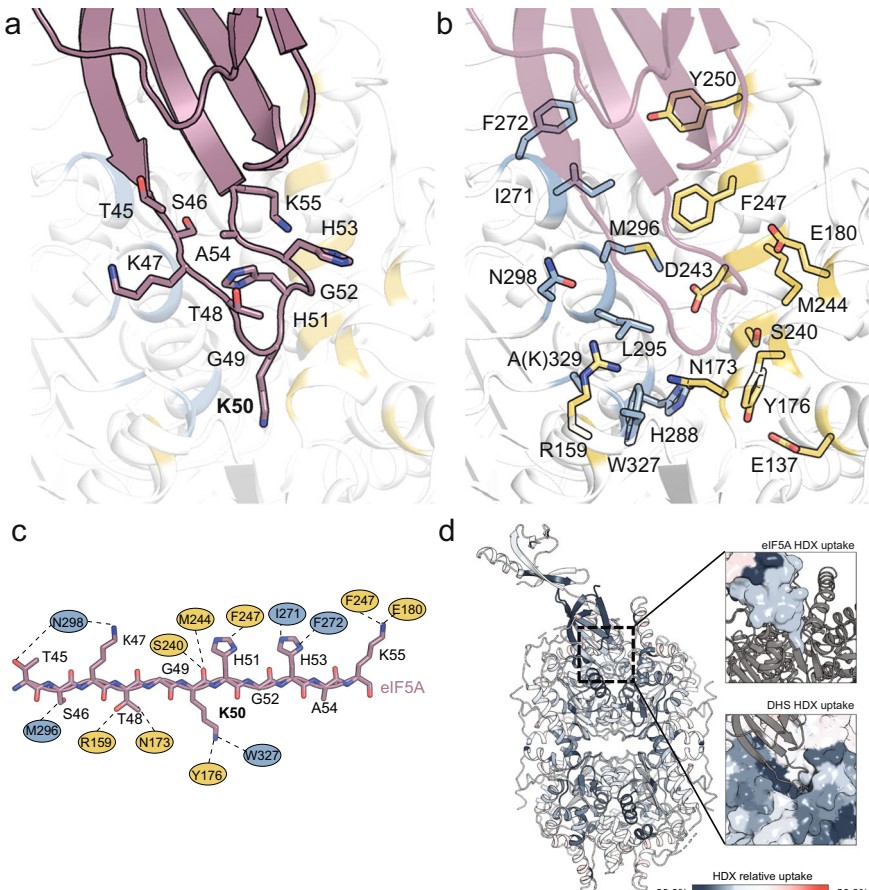

**Fig. 2 | Details of the interaction architecture.** Visualization of the eIF5A-DHS binding site (**a**, **b**). Side chains involved in the interaction are shown as stick representations (a and b for eIF5A and DHS, respectively). **c** Simplified visualization of the eIF5A hypusination loop depicted as an extended peptide (dark pink) with marked interacting residues of DHS monomer A (blue) and B (yellow). **d** The eIF5A-DHS complex in cartoon representation is colored according to the result of the HDX-MS experiment. Inlets present eIF5A (top) or DHS (bottom) in surface representation. Bluish coloring represents reduced deuterium incorporation of the eIF5A-DHS complex compared to the individual proteins reflecting alterations in protein conformation and shielding from bulk solvent.

(Supplementary Fig. 6). In contrast, the difference in the portions of fast and slow-exchanging species between eIF5A and eIF5A-DHS$^{K329A}$ peptides from the C-terminal part of eIF5A (residues 90–101, 108–131, 111–134) was less severe indicating that the conformational flexibility in this part depends less on the presence of DHS (Supplementary Fig. 6). These results may also explain the weakly defined cryo-EM density of eIF5A, in particular in its C-terminal portion.

DHS exhibited perturbations in HDX in multiple parts of the protein. In particular, the eIF5A binding pocket of DHS (residues 104–118, 129–183, 266–278, and 282–295) and the active site loop (residues 306–344) showed HDX reductions (Fig. 3a, Supplementary Fig. 4). Additionally, increased HDX in residues 37–42 and 355–361, and a decrease in 49–62, regions that are remote from the eIF5A-binding and active sites, suggest alterations in the overall topology of the DHS homotetramer.

In almost all areas of DHS where the protein incorporated less deuterium in the context of the eIF5A-DHS complex, bimodal HDX behaviour was apparent that was characterized by a higher portion of the slow-exchanging population in eIF5A-DHS than in DHS alone, e.g. in peptides spanning residues 139-151, 152-169, 263-272, 279-293 and 312-327 (Supplementary Figs. 5d–f, Supplementary Fig. 7). One weakness of HDX-MS is that this method cannot discriminate between eIF5A-bound and empty DHS molecules. In this regard, the observed bimodality may simply reflect these different DHS species, which are a consequence of the asymmetry of the complex. However, bimodal HDX behaviour was also apparent for DHS in absence of eIF5A, and a peptide containing residues 47–59, which is remote from the eIF5A

binding site. This suggests that DHS intrinsically exhibits a certain degree of conformational flexibility in the absence of eIF5A, and alterations in its complex topology upon eIF5A binding.

Collectively, the HDX-MS data corroborate the features of the DHS-eIF5A interaction inferred from the cryo-EM data in solution and furthermore identify subtle alterations in the tetrameric topology of DHS that occur upon eIF5A binding.

### Structure-guided mutagenesis separates DHS catalytic activity and eIF5A binding

To test the functional significance of the interactions defined by our structural work, we selected and mutated several residues from the eIF5A-DHS interface (Gln83, Asn173, Tyr176, Ser240, Phe247, Tyr250, Ile271, Phe272, Leu295) and the active site of DHS (Glu137, Asp243, His288, Trp327, Lys329). After the production and purification of the variants (see "Methods"), we analyzed their contribution to stability, activity, and complex formation (Fig. 3).

First, we assessed the thermostability of the individual proteins using differential scanning fluorimetry (DSF). Most of the mutations analyzed do not or only slightly affect the stability of DHS, but the mutation of hydrophobic interface residue (I271A) showed significantly decreased melting temperature (Fig. 3a).

We next examined catalytic activity. As mentioned above, DHS catalyzes the transfer of the 4-aminobutyl moiety from SPD to the modifiable lysine (Lys50) of eIF5A in a NAD-dependent manner. The reduction of NAD, when a transient DHS-NADH-SPD complex is formed, can be assessed by recording the intrinsic fluorescence of

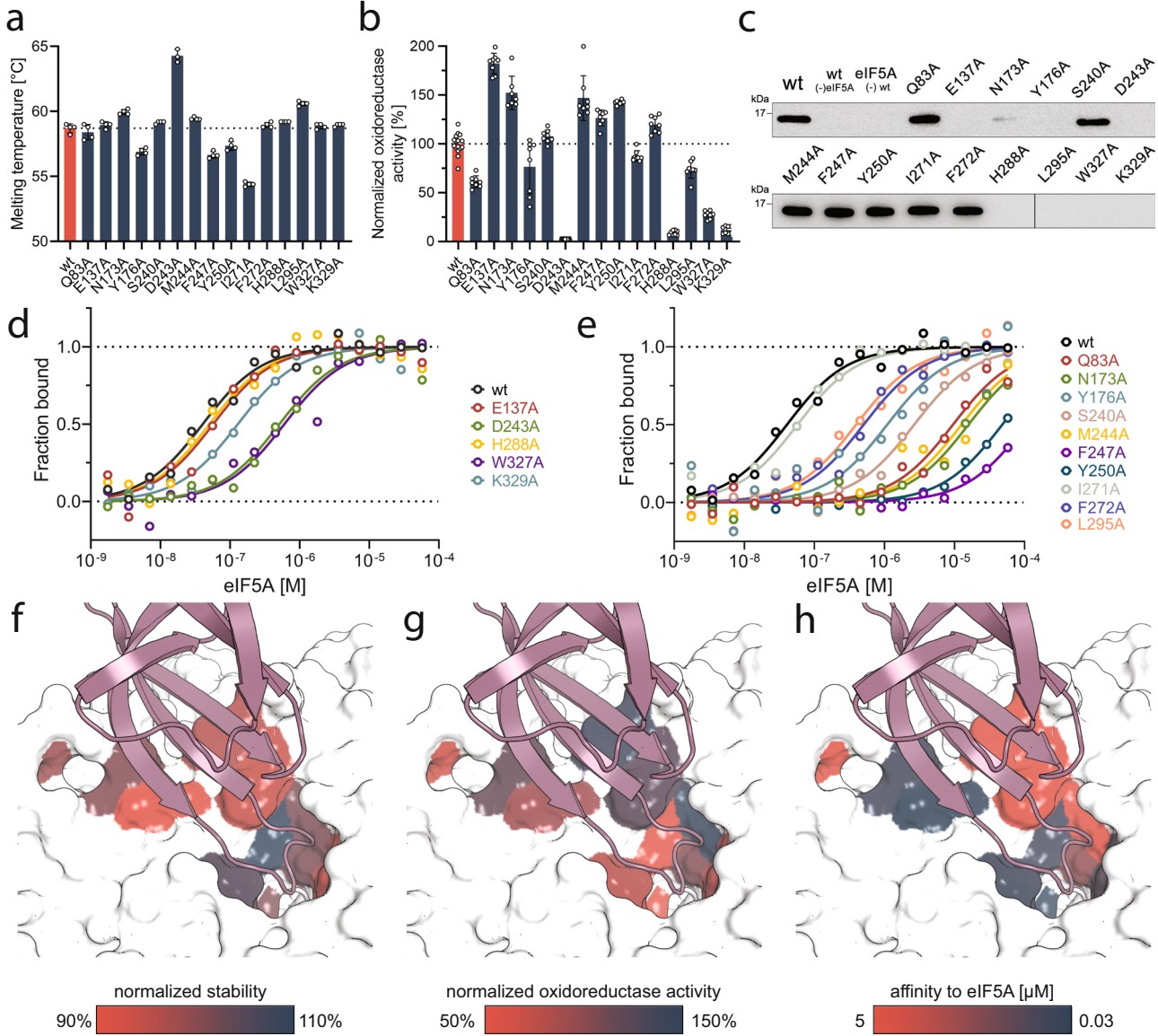

**Fig. 3 | Site-directed mutagenesis studies.** The significance of selected interface or active site residues of DHS was demonstrated by their alanine substitution. The influence on various aspects is shown in consecutive panels: Thermal stability (**a**), oxidoreductase activity (**b**), hypusination capacity (**c**), affinity to eIF5A of the active site (**d**) and interface substitutions (**e**). For the thermal stability (**a**) and oxidoreductase activity (**b**) assays, data were normalized relative to wild-type. Bars represent the mean ± SD of at *n* = 4 (**a**) and *n* ≥ 7 (**b**) independent experiments. Dots represent individual measurements. In hypusination assay (**c**) for deoxyhypusination detection, monoclonal rabbit FabHpu98 antibody was used[52]. See the "Methods" section for details of assays. Representative blots of three independent experiments are shown. Panels **f**–**h** map the strength of the effect in color-coded visualization with the scale indicated below each panel. Source data for all panels are provided in Source Data file.

NADH[29]. We used this assay to analyze the oxidoreductase activity of the DHS variants. Based on the structure, we expected that mutations of the complex interface would rather not affect the DHS oxidoreductase activity. Not surprisingly, mutations in active site residues (D243A, H288A, W327A, and K329A) resulted in almost complete suppression of SPD dehydrogenation activity. A decrease in activity was also observed for some mutations of the complex interface (Q83A, I271A, and L295A). The opposite results were obtained for alanine mutants of residues mapping to the interface between eIF5A and DHS (N173A, S240A, M244A, F247A, Y250A and F272A) and active site mutant E137A. These mutations increased oxidoreductase activity (Fig. 3b). This result may reflect higher solvent accessibility, which may influence the NADH hydration state.

To further validate the importance of the above-mentioned residues in the catalytic reaction, we performed a qualitative hypusination assay to directly measure modification activity. As expected, mutations introduced in the active site of DHS (E137A, D243A, H288A, W327A, K329A), as well as three interface mutants directly involved in the stabilization of the eIF5A hypusination loop (N173A, Y176A, and L295A), completely inhibited hypusination activity. The remaining mutations did not affect the hypusination efficiency (Fig. 3c).

To elucidate the role of selected residues in complex formation, we measured the binding affinity between eIF5A and DHS using microscale thermophoresis (MST). For DHS$^{wt}$ the binding affinity was in the high nanomolar range: 33.8 ± 8.5 nM. The mutation in active site residue H288A did not affect the eIF5A binding ($K_D$ 49.7 ± 28.7 nM). On the other hand, the K329A substitution showed a decrease in affinity ($K_D$ 128.6 ± 37.9 nM; Fig. 3d, Supplementary Fig. 8). Moreover, mutation of the Trp327 residue caused a significant decrease in affinity to 602.5 ± 284.2 nM, highlighting the importance of this residue for

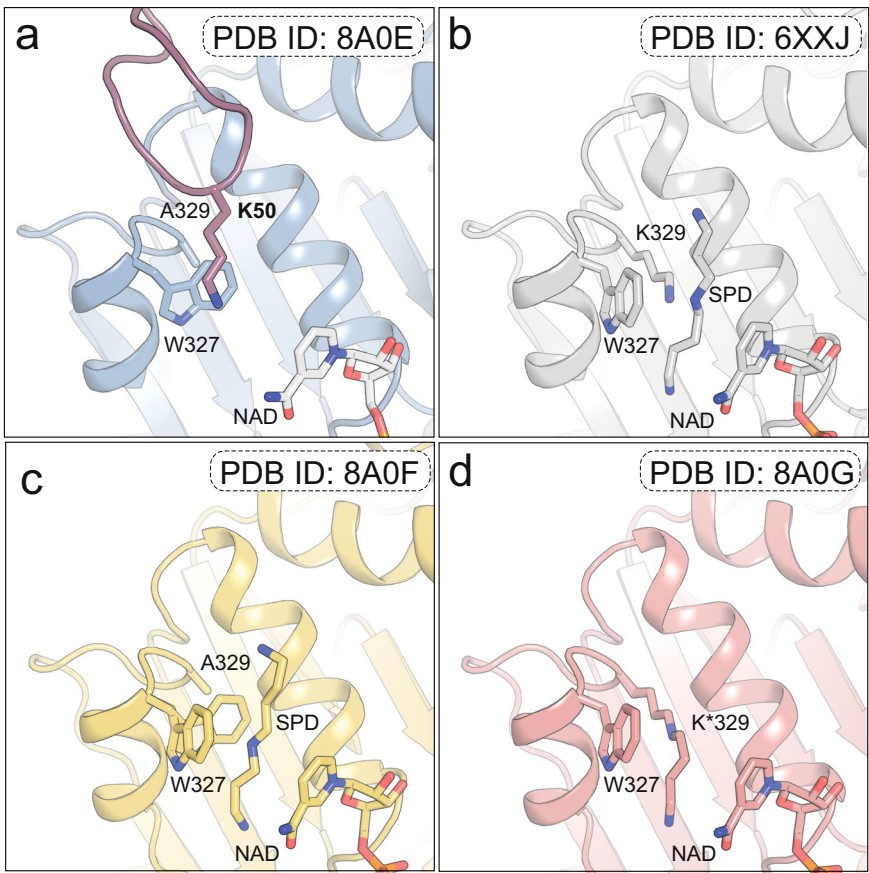

**Fig. 4 | DHS active sites comparison.** Comparison of the architecture of the active site of **a** the eIF5A-DHS$^{K329A}$ complex (8A0E), **b** DHS wt (6XXJ), **c** DHS$^{K329A}$ in complex with NAD and SPD (8A0F) and **d** DHS with the intermediate state on lysine 329 (8A0G). Ligands and key residues are shown as sticks.

complex formation. A similar effect was observed for D243A (corresponding to the SPD binding site) with an increased $K_D$ value of $511.3 \pm 182.1$ nM, and for E137A (NAD binding site) with a $K_D > 1\,\mu$M. All residues at the eIF5A-DHS interface contribute greatly to complex formation, and their alanine mutants exhibit an almost complete diminishment of the recruitment of eIF5A (Fig. 3e, Supplementary Fig. 8). These results underline the idea that hydrophobic interactions play a substantial role and that substitutions within the catalytic center of DHS (e.g. K329A) do not abolish the interaction with eIF5A. Interestingly, results from the activity and binding assays show that residues crucial for proper eIF5A binding are not essential for oxidoreductase activity (Fig. 3f–h) and vice versa.

**Visualization of a trapped reaction intermediate during DHS catalysis**

The results from our structure-guided analyses suggested that the binding of eIF5A unblocks the active site. To examine whether eIF5A binding induces additional changes at the active site, we compared our eIF5A-DHS structure with the previously solved structure of the wild-type DHS protein (DHS$^{wt}$)[16]. Interestingly, we observed that the indole ring of Trp327 is shifted perpendicularly in the complex, relative to its position in DHS$^{wt}$ (Fig. 4a, b). This rearrangement of the side chain caught our attention because the Trp327 residue is involved in the hydrophobic stabilization of the SPD methylene groups[18,30]. The cryo-EM structure clearly indicates that the wild-type position of Trp327 overlaps with that of the modifiable Lys50 residue of eIF5A in the complex. Simultaneously, within DHS, the rearranged position of Trp327 would collide with Lys329, which was mutated to alanine in the trapped complex used in our structural studies. It was therefore plausible that this conformational change was a direct or indirect

result of the K329A mutation, rather than eIF5A binding per se. To address this issue, we determined the crystal structure of DHS$^{K329A}$ in complex with NAD and SPD and refined it at 1.6 Å (Supplementary Table 3). DHS$^{K329A}$ crystallized in the P3$_2$21 space group with one tightly associated DHS dimer per asymmetric unit. The DHS tetramer, which constitutes the active form of the enzyme, is completed by the neighboring crystallographic symmetry mate. In the structure of DHS$^{K329A}$-NAD-SPD, we observed two alternative conformations of the Trp327 side chain with almost equivalent occupancies. In detail, we found the perpendicular position that is also found in the eIF5A-DHS complex and the position that is parallel to the bound SPD molecule, as in DHSwt (without protein ligand) (Fig. 4c). The observed conformational change indicates that the K329A mutation gives rise to greater conformational freedom of the bulky indole chain of Trp327.

Given the above results, we wondered how the conformational change involving the Trp327 indole chain impacts the dexyhypusination reaction. In the early stage of the reaction catalyzed by DHS, SPD is cleaved to the 1,3-diaminopropane (DAP) and the 4-aminobutyl moiety. 4-aminobutyl subsequently forms an imine linkage to Nε of Lys329 in DHS, resulting in a transient enzyme-imine product intermediate[31]. This transient double bond intermediate can be reduced to a stable single-bond in the presence of NaBH$_3$CN[32], leading to a covalent modification and the accumulation of the transition state analogue, namely deoxyhypusinated Lys329, and DAP (Supplementary Fig. 9a). To test this biochemical model, we sought to trap and visualize the reaction transition state. To do so, we collected a diffraction dataset of a DHS-NAD-SPD crystal pretreated with NaBH$_3$CN at room temperature (RT). The intermediate state structure of DHS was solved with the same packing scheme and space group as the other DHS crystal structures and was refined at 1.8 Å resolution (Supplementary Table 3).

As expected, the overall structure of the DHS reaction transition state is like those of previously reported DHS structures. However, our soaking approach at RT allowed us to observe a continuous electron density extending the Lys329 side chain, suggesting that this residue has been modified to deoxyhypusine upon $NaBH_3CN$ treatment (Fig. 4d, Supplementary Fig. 9b, c). The 4-aminobutyl moiety is covalently bound to the ε-amine group of Lys329 and stabilized by the Trp329 indole and NAD pyridine rings. Moreover, the electron density observed close to the exit of the active site tunnel likely represents DAP. These analyses therefore demonstrate the existence of the proposed intermediate and suggest that the rearrangement of the Trp327 would lead to a collision with the deoxyhypusinated Lys329 residue. Thus, the spatial restriction of the Trp327 indole ring in the wild-type DHS protein may facilitate the transfer of 4-aminobutyl moiety from Lys329 to Lys50 of eIF5A.

### Pathological mutations impact DHS stability and function

Recently, a set of clinically relevant mutations in the *dhps* gene were identified in five patients with neurodegenerative disease[22] (Supplementary Fig. 10a, b). All patients carry an N173S mutation along with a secondary mutation. As the above data provided insight into DHS domains that may be critical for function, we sought to understand how disease-associated changes may impact DHS function. We chose to focus on DHS[N173S] and one of the secondary mutations, DHS[delY305_I306] as only those variants can be potentially expressed as active proteins in the cell[22]. Asn173 is part of the helix flanking the entrance tunnel to the active site. It interacts with the side chain of the neighboring Asn164 and with the Val17 and Lys19 main chains of the *ball-and-chain* motif (Supplementary Fig. 10b). The residues absent in DHS[delY305_I306] are located within the protein core, in one of the β-strands of the Rossman fold. These residues assist in the stabilization of the NAD binding pocket through an extensive interaction network of the main chain.

We expressed and purified both variants (see "Methods"). Initial trials showed a significant decrease in the solubility of DHS[delY305_I306], indicating possible misfolding or aggregation of the protein. However, a small portion of DHS[del305Y_306I] remained soluble and we were able to obtain enough purified protein for further experiments. Of note, solubilization and refolding procedures were employed previously to assess the behavior of both DHS variants in vitro[22], but we purified all DHS variants directly from the soluble fraction. As above, we first assessed protein stability using DSF. Surprisingly, the melting temperature ($T_m$) of the DHS[N173S] variant (64.1 ± 0.6 °C) is comparable with that of the wild-type protein (DHS[wt]) (62.9 ± 0.1 °C), whereas the $T_m$-values observed for DHS[del305Y_306I] were significantly lower (46.7 ± 0.1 °C; Fig. 5a, Supplementary Fig. 10c). Next, we analyzed the oligomerization potential of the mutant proteins by mass photometry, which revealed differences in the oligomeric states of DHS variants. Whereas the profile of DHS[N173S] resembles the wild-type tetrameric form, the DHS[del305Y_306I] variant is primarily dimeric in solution (Fig. 5b).

### Structural analysis of the DHS[N173S] variant

Given the weaker effects of the N173S mutation, we pursued structural studies to better understand its impact on DHS function. We solved a high-resolution structure of DHS[N173S] in a complex with NAD and SPD. Despite intensive efforts, we were unable to crystallize the DHS[del305Y_306I] variant. DHS[N173S] crystallized in the P3₂21 space group with one tightly associated dimer per asymmetric unit, related to an identical dimer by a twofold axis. The structure was refined at 1.7 Å resolution (Supplementary Tab. 3). Structural comparison between wild-type and mutant DHS showed only minor differences of individual monomers (RMSD 0.15 Å) and the whole tetramer (RMSD 2.2 Å; Fig. 5c). The only significant difference is found near the mutated site. As mentioned above, Asn173 is located on the surface of the protein and stabilizes the neighboring *ball-and-chain* motif via polar interactions (with residues Val17, Lys19). The mutated serine residue

in DHS[N173S] maintains the interaction with Asn164, but its side chain, having only one polar hydroxyl group, cannot form hydrogen bonds with the N-terminal part of the proximal protomer (Fig. 5d, e). These lost interactions may underlie the lack of observable electron density for the *ball-and-chain* motif, indicating poor ordering and partial destabilization of this motif in the variant structure. Mapping the N173S mutation onto the structure of the eIF5A-DHS complex suggested that the DHS[N173S] variant would also disturb the positioning of eIF5A in the DHS active site. In the structure of the eIF5A-DHS complex, the Asn173 residue of DHS interacts with the main chain of the eIF5A Thr48, constituting part of the interface of the complex (Supplementary Fig. 10d–f).

Structural analysis of DHS[N173S], therefore, confirms that the mutation does not have a significant impact on overall protein structure but does affect the *ball-and-chain* motif, and likely also impacts the interaction with eIF5A.

### Pathological DHS mutations impair its activity and ligand binding

Next, we employed single-turnover fluorescence assays[33], monitoring SPD dehydrogenation, to assess the influence of the mutations on enzymatic activity. An excess of SPD in the reaction triggers a rapid increase of fluorescence signal for both the wild-type and the N173S variant proteins. Interestingly, the final value of the fluorescence signal is ~25% higher for the N173S variant, but the progress of the reaction is identical for both wt and N173S variants (Fig. 5f). Concomitantly, the DHS[305Y_306I] variant was inactive and no readout was observed. The higher fluorescence values for the N173S variant may reflect increased oxidoreductase activity but may also reflect differences in the hydration state of NADH, indirectly triggered by the destabilization of the *ball-and-chain* motif[4]. Examining the hypusination reaction, the N173S variant led to a lower level of detected deoxyhypusine (Fig. 5i), while the deletion mutant yielded no deoxyhypusine product at all.

We also examined the impact of pathological mutations on SPD binding using a FRET assay (Fig. 5g)[29]. The calculated apparent $K_D$ values for SPD were 3.3 ± 0.3 (wt) and 27.0 ± 3.1 µM (N173S), respectively (Fig. 5h), indicating reduced binding. In the case of DHS[del305Y_306I], we did not observe any binding. The lack of binding is most likely caused by a general structural destabilization of this mutant.

Finally, we also investigated the impact of pathological mutations of DHS on the binding to eIF5A using MST assays. The $K_D$ values for the wild-type proteins are 33.8 ± 8.5 nM. The DHS variants show decreased binding parameters (115.1 ± 22.3 nM for N173S and 80.0 ± 23.4 nM for del305Y_306I), respectively. Thus, our results indicate a loss in affinity for eIF5A for both patient-derived DHS variants (Fig. 5j). The addition of NAD cofactor to the assay mixture did not change the strength of the interaction in the case of the wild-type and N173S variant proteins, whereas del305Y_306I showed a significant loss of binding affinity (Fig. 5k). In the presence of SPD we observed a further reduction in binding affinity for all three proteins (Fig. 5l). These results suggest insignificant changes within the core of the protein in the case of N173S and imply that the del305Y_306I deletion does not fully prevent NAD binding.

Together, the biochemical and mutational analyses suggest strong impacts of the del305_306I deletion on DHS stability, enzymatic activity, and substrate binding. In the case of the N173S mutation, the variant protein is stable, and the effects on activity and substrate binding are less pronounced.

## Discussion

Hypusination of eIF5A is an essential modification in eukaryotic cells. Impairment in this process leads to severe human disorders. Here, we employ high-resolution cryo-EM studies to examine the mechanism underlying the formation of the eIF5A-DHS complex. Furthermore, we visualize the previously hypothesized deoxyhypusination reaction

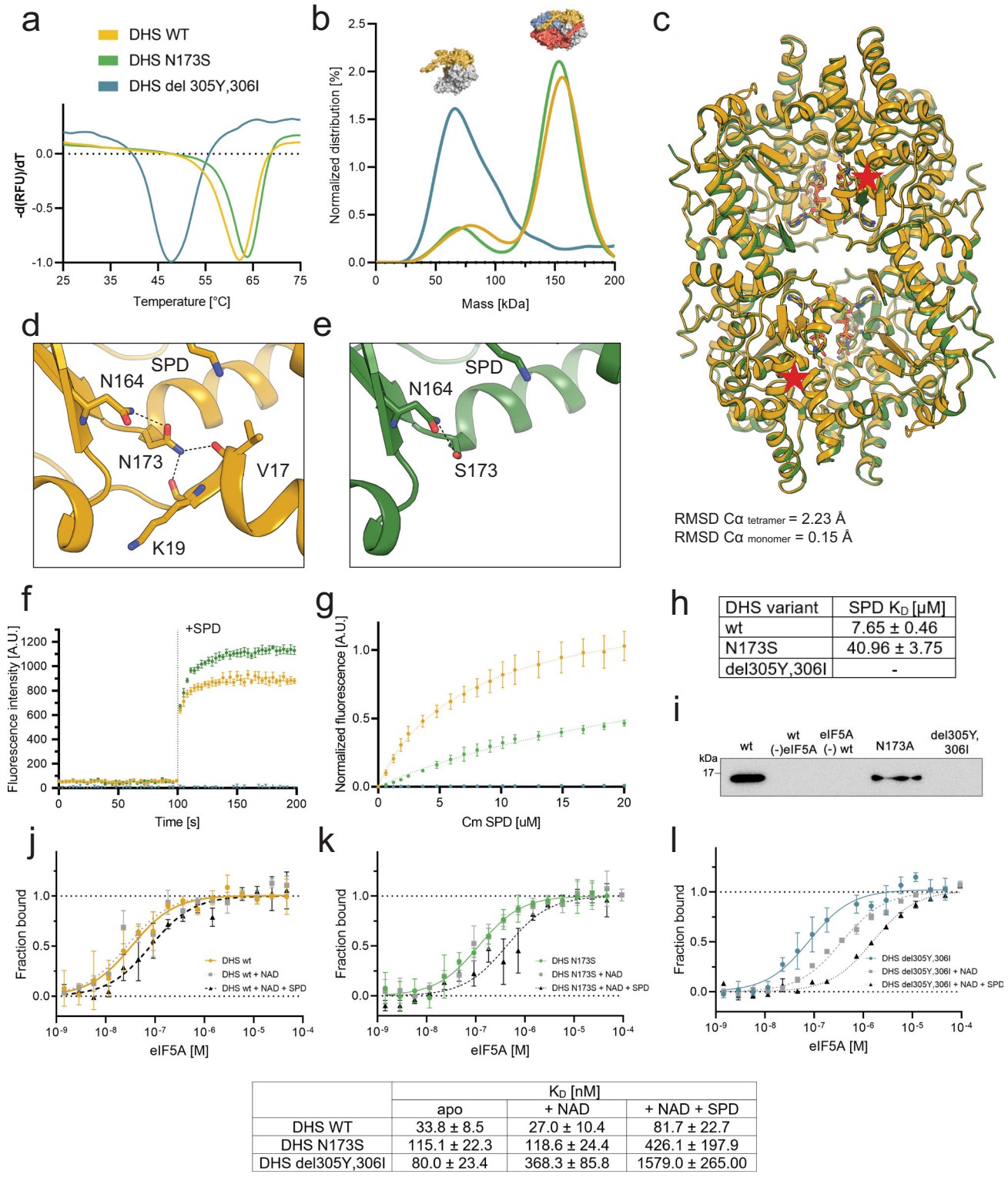

| DHS variant | SPD $K_D$ [µM] |
|---|---|
| wt | 7.65 ± 0.46 |
| N173S | 40.96 ± 3.75 |
| del305Y,306I | - |

RMSD Cα $_{tetramer}$ = 2.23 Å
RMSD Cα $_{monomer}$ = 0.15 Å

| | $K_D$ [nM] | | |
|---|---|---|---|
| | apo | + NAD | + NAD + SPD |
| DHS WT | 33.8 ± 8.5 | 27.0 ± 10.4 | 81.7 ± 22.7 |
| DHS N173S | 115.1 ± 22.3 | 118.6 ± 24.4 | 426.1 ± 197.9 |
| DHS del305Y,306I | 80.0 ± 23.4 | 368.3 ± 85.8 | 1579.0 ± 265.00 |

transition state using ambient-temperature X-ray crystallography. Finally, we unveil the multifaceted influence of clinically relevant DHS mutants on the hypusination reaction.

Previous studies reported the crystal and cryo-EM structures of eIF5A bound to the ribosome[20,34]. The mechanism of eIF5A binding and export from the nucleus to the cytoplasm, using the Xpo4 and Pdr6 exportin systems, has also been elucidated[35,36]. The structures described herein allow us to complete the molecular landscape of eIF5A recognition by its binding partners. Our studies show that DHS binding utilizes only the part of the N-terminal domain of eIF5A containing the β-sheet structure. In contrast, the interaction of eIF5A with the ribosome is much more complex, involving almost the complete N-terminal and part of C-terminal domains (Fig. 6). In this context it is interesting to note that, of the recently described clinical variants of eIF5A[37], only one (a T48N substitution) maps to the DHS binding site in eIF5A, whereas the rest affect ribosome binding sites. Furthermore, the binding sites of eIF5A to exportins also involve a greater number of residues than those that mediate the interaction with DHS. In the case

**Fig. 5 | Structural and biochemical analysis of loss-of-function DHS variants.**
**a** Thermal stability of DHS variants assessed by DSF. The color-coding indicated above panel a is applied to subsequent panels as well. **b** Mass photometry profiles obtained for DHS wt (yellow), DHS^NI73S (green), and DHS^del305Y,306I (blue) with the determined oligomeric state indicated as a schematic surface representation above the respective peaks. **c** Superposition of structures of wild type (PDB ID: 6XXJ) (yellow) and DHS^NI73S (PDB ID: 7A6T) (green). The mutation sites on the front of the molecules are indicated by a red star. The similarity for monomers or entire tetramers is expressed as the root-mean-square deviation (RMSD) for the corresponding Cα atoms and is indicated below the image. **d**, **e** Comparison of the bonding pattern of residue 173 in DHS wt (yellow) and DHS^NI73S (green). The main chain is shown as a cartoon representation. The mutant residues and the interacting residues are shown as sticks. Electrostatic interactions are shown as black dashed lines. **f** Single-turnover fluorescence activity assay performed for DHS wt and

pathological variants. On the graph mean values ± SD of $n \geq 3$ independent experiments are presented. **g** FRET-based binding assay with normalized fluorescence plotted as a function of SPD concentration and **h** the derived dissociation constant values tabularized. On the graph mean values ± SD of $n = 3$ independent experiments are presented. **i** Hypusination assessed by western blot. For deoxyhypusination detection, monoclonal rabbit FabHpu98 antibody was used[52]. The variant used is indicated above each line. Representative blots of three independent experiments are shown. **j–l** The affinity of DHS^wt, DHS^NI73S and DHS^del305Y,306I, respectively, for eIF5A was assessed using MST, either in free buffer, with cofactor, or with cofactor and substrate supplemented as indicated by the legend. Data represent the mean ± SD of $n = 3$ individual experiments. The numerical values derived from the experiments are tabularized below. Source data for all panels are provided in Source Data file.

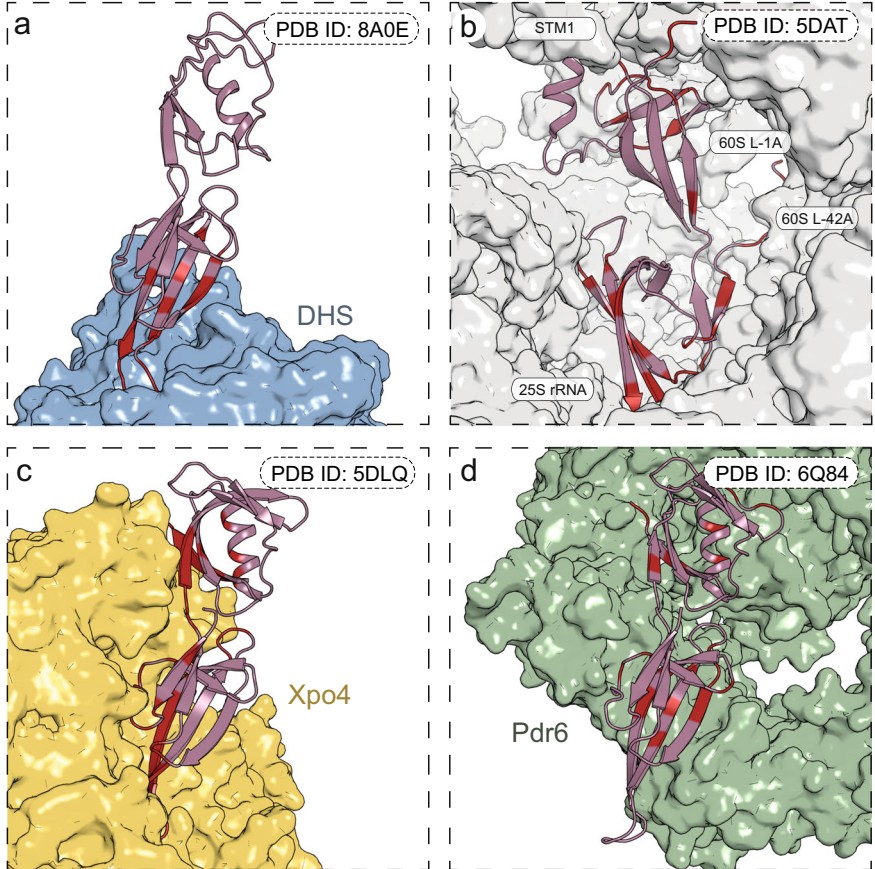

**Fig. 6 | Comparison of eIF5A interfaces.** Several complexes bearing eIF5A are present in the PDB allowing us to compare the proteins binding interfaces responsible for various functionalities including posttranslational modification, nuclear import/export and resolving ribosomal stalling. eIF5A is shown as a dark pink cartoon in a complex with a binding partner shown as a coloured surface. Contact points, i.e. eIF5A in proximity less than 4.5 Å of the binding partner are marked with red. Depicted are complexes with **a** DHS (PDB ID: 8A0E blue), **b** ribosome (PDB ID: 5DAT, white), **c** Exportin-4 (PDB ID: 5DLQ, yellow) and **d** importin protein KAP122 (PDB ID: 6Q84, green). eIF5A is shown in the same relative position except for the ribosome where it is rotated by 180 degrees for the sake of clarity.

of interaction with Xpo4, exportin recognizes both globular domains of eIF5A and the hypusine residue is buried in a deep acidic binding pocket. On the other hand, the binding of eIF5A to Pdr6 involves a smaller part of the N-terminal domain not containing the hypusination loop. Thus, eIF5A-DHS and eIF5-Pdr6 overlap only partially.

Our cryo-EM and crystal structures allow us to gain insights into the molecular mechanisms governing the synthesis of deoxyhypusine and to propose a highly plausible visualization of the reaction (Supplementary Movie 1). In the early stage of the DHS-catalyzed reaction, SPD is oxidized and cleaved into 1,3-diaminopropane (DAP) and the 4-aminobutyl moiety, which subsequently forms an imine linkage to Nε of K329^DHS. This reaction is coupled with the reduction of NAD to

NADH. As shown in the DHS transition-state mimic crystal structure, the SPD N6 atom is sequestered from the reduced nicotinamide of NADH. Hence, the NADH-driven reduction of the transition-state imine bond is significantly attenuated. Further analysis of the eIF5A-DHS structure highlights the importance of W327^DHS, which serves as a gating residue for the incoming K50 of eIF5A. The conformational change of the W327^DHS side chain upon binding to eIF5A is not only essential for the proper positioning of the to-be-hypusinated K50^eIF5A, but it may also induce a significant shift of K329^DHS, allowing the 4-aminobutyl transfer to form an imine intermediate with K50^eIF5A. In turn, the imine bound becomes accessible to the hydride ion transferred from the NADH and deoxyhypusine (Dhp50^eIF5A) is formed.

Our findings also provide a framework for a better understanding of the molecular mechanism of DHS deficiency. The reported exome sequencing of individuals affected with this recently recognized rare hereditary disease identified a total number of four *dhps* variants. However, each of the affected individuals has a single copy of the DHS[N173S] variant. Thus, we and others hypothesized that the N173S variant retained partial activity allowing for the survival and further organismal development. Indeed, we find that the N173S retains overall structural integrity and some enzymatic activity. However, our data also suggest that the N173S substitution influences the stability and/or mobility of the *ball-and-chain* motif of DHS. This, in turn, may affect the accessibility of the active site to the substrates, namely spermidine and eIF5A. Furthermore, N173[DHS] not only forms a hydrogen bond with another active site residue (N164[DHS]) but also interacts with T48[eIF5A]. As mentioned above, the substitution of this residue has been associated with neurological disease[37]. Indeed, mutations affecting both N173[DHS] and T48[eIF5A], lead to similar clinical manifestations, consistent with our structural studies mapping both of these residues to the interface between eIF5A and DHS.

Taken together, we describe the molecular basis for eIF5A recognition by DHS and broaden our understanding of the hypusination reaction. Furthermore, we reveal the molecular mechanisms underlying DHS deficiency, the recently described rare hereditary disease. The importance of the hypusination axis for neural development has been demonstrated in a recent study showing that neuron-specific ablation of eIF5A or DHS leads to impairments in neurodevelopment and cognitive functions in mice[38]. Individuals with certain variants of either eIF5A[37] or DOHH[39] display similar developmental delay and intellectual disability to those affected by DHS deficiency[22]. Hence, future research needs to investigate the effects of those mutations on the mechanisms of hypusination.

## Methods

### Protein cloning, expression and purification

The *dhps* gene encoding for full-length *Homo sapiens* (Uniprot: P49366, residues 1–369) and two pathological variants: N173S and del305Y_306I were synthesized (Genescript) and cloned with an N-terminal 6xHis-tag followed by a TEV cleavage site into pET24d plasmid using NcoI/BamHI restriction sites. For eIF5A expression pET28-MHL (Addgene) encoding 15-151 sequence of eIF5A1 fused to an N-terminal HisTag followed by a TEV recognition site or pETM-40 encoding eIF5A-2 with N-terminal MBP tag were used. Additionally, the pET24d plasmid encoding full-length DHS wt was used as a template for site-directed mutagenesis. Mutation specific primers were designed and purchased from Merck. Point mutations were introduced with site-specific primers (Merck) by PCR reaction and the DpnI-treated PCR products were used to transform XL1-Blue super-competent cells, which were grown on agar plates in the presence of kanamycin. The presence of appropriate mutations was confirmed by DNA sequencing (Eurofins Genomics GmbH). Human eIF5A1 and all DHS variants were expressed in *Escherichia coli* and purified according to the previously described procedures[16]. In short, cells were collected by centrifugation (17,700 × *g*, 12 min, 4 °C) and re-suspended in lysis buffer (50 mM Tris-HCl pH 7.8, 300 mM NaCl, 20 mM imidazole, 10% glycerol, 10 mM β-ME). Cells were disrupted in the presence of lysozyme (Sigma-Aldrich) by sonication (15 min, 5 s pulse/3 s pause cycles). Homogenous cell solution, with the addition of benzonase (Sigma-Aldrich), was subjected to centrifugation (53,000 × *g*, 45 min, 4 °C). After centrifugation, the cleared lysate was applied onto an equilibrated affinity column (5 mL HisTrap HP; GE Healthcare Europe GmbH, Freiburg, Germany) and washed with wash buffer (50 mM Tris-HCl pH 7.8, 200 mM NaCl, 40 mM imidazole, 5% glycerol, 5 mM β-ME) to elute non-specifically bounded proteins. The protein of interest was eluted with the elution buffer (50 mM Tris-HCl pH 7.8, 200 mM NaCl, 400 mM imidazole, 5% glycerol, 5 mM β-ME). Eluted protein was dialyzed

against the storage buffer (50 mM Tris-HCl pH 7.8, 200 mM NaCl, 5 mM β-ME) and subjected to overnight TEV protease cleavage during the second step of dialysis to remove the affinity tag. After HisTag cleavage, the tag-free protein was separated from the undigested protein and HisTagged-TEV protease during the reverse HisTrap column chromatography. Fractions containing HisTag-free protein were concentrated using Amicon Ultra (Millipore) concentrator (cut-off: 10,000 kDa) and subjected to size-exclusion chromatography (SEC) on a HiLoad 16/60 Superdex 75 column in a storage buffer. Peaks of the highest purity were pooled, concentrated, aliquoted and flash-frozen in liquid nitrogen for further analysis. The full-length eIF5A-2 (Uniprot: Q9GZV4, residues 1–153) was synthesized (Genescript), and cloned into pETM40 vector using NcoI/XhoI restriction sites and expressed as an MBP-fusion protein in *E. coli* BL21(DE3) cells. Protein was purified using affinity (5 mL MBP-Trap HP; GE Healthcare Europe GmbH, Freiburg, Germany) and size-exclusion chromatography in storage buffer as an MBP-eIF5A-2 fusion protein.

### Complex reconstitution

We reconstituted the human eIF5A-DHS[K329A] complex using previously purified individual components namely eIF5A-1 and catalytically inactive DHS[K329A]. DHS[K329A] was mixed with a molar excess of eIF5A-1 (2 mole eIF5A per 1 mole DHS). Proteins were incubated in 200 mM Glycine/NaOH pH = 9.3 buffer with 200 mM NaCl in the presence of 1 mM NAD and 1 mM SPD. The complex was then purified by size-exclusion chromatogram using a Superdex 200 Increase 10/300 column (GE Healthcare) pre-equilibrated with 200 mM Glycine/NaOH pH=9.3 buffer with 200 mM NaCl. Due to the very small difference in mass of the complex relative to DHS alone the fraction from the very front of the elution peak was used for further investigation.

### Protein crystallization

DHS[wt], DHS[N173S] and DHS[K329A] crystals were obtained using previously determined conditions[16]. Briefly, 0.5 μl of protein (~20 mg/ml) was mixed with an equal volume of mother liquor solution consisting of a 0.025–0.125 mM carboxylic acid mix, 30–60% precipitant mix (MPD, PEG 1000, PEG 3350) and 100 mM Tris-Bicine with a pH of 8.5 and equilibrated by a sitting drop vapour diffusion technique. To obtain crystals in a complex with NAD and SPD, an equal volume of 50 mM mixture of SPD and NAD was added to the crystallization drop. Crystals appeared after 2 days and were soaked in cryo-solution containing 25% of ethylene glycol in mother liquor and flash frozen in liquid nitrogen. For trapping the intermediate state a crystal was incubated for approximately half a minute in the well-solution supplemented with 10 mM NaBH$_3$CN directly before mounting on the goniometer.

### Diffraction data collection and structure determination

Diffraction data for DHS[N173S] and DHS[K329A] crystals were collected at the MX-beamline 14.1 in cryogenic conditions. Diffraction data for the intermediate state were collected at MX-beamline 14.3 at room temperature under controlled humidity. Both beamlines are operated at the BESSY II electron storage ring (HZB, Berlin, Germany)[40]. Diffraction data were processed using XDS as implemented in the XDSAPP3 v.1.8 package[41]. All DHS crystal structures were solved by molecular replacement with Phaser[42] using 6XXJ as a search model and rebuilt using *Coot*[43]. ~1.5% of the reflections were used for cross-validation analysis to monitor the refinement strategy in Phenix[5]. Water molecules were automatically placed during structure refinement, or further added using Coot and subsequently manually inspected. The quality of the model was validated using MolProbity[44] and the final resolution cut-off was applied according to the PAIREF software[45]. The crystal structures were finalized with satisfactory geometrical parameters and rather low $R_{work}/R_{free}$ values indicating the good quality of the structure. All significant data collection, structure refinements, and validation statistics are summarized in Supplementary Table 3. The analysis and

comparison of structures were performed in PyMOL (Molecular Graphics System, Version 2.0 Schrödinger, LLC) or UCSF Chimera[46].

## Sample preparation and cryo-EM data collection

Approximately 3 μL of the sample solution was applied on freshly glow-discharged TEM grids (Quantifoil R2/1, Cu, mesh 200) and plunge-frozen in liquid ethane with the use of Vitrobot Mark IV (Thermo Fisher Scientific). The following parameters were set− humidity: 100%, temperature: 4 °C, blot time: 2 s. Frozen grids were kept in liquid nitrogen until clipping and loading into the microscope. Cryo-EM data were collected at National Cryo-EM Centre SOLARIS (Kraków, Poland). Datasets containing 12419 movies (40 frames each) were collected with Titan Krios G3i microscope (Thermo Fisher Scientific) at the accelerating voltage of 300 kV, magnification of 105k and corresponding pixel size of 0.86 Å/px. K3 direct electron detector was used for data collection in BioQuantum Imaging Filter (Gatan) setup with 20 eV slit enabled. K3 detector was operated in counting mode with physical pixel resolution. Imaged areas were exposed to 39.90 e-/Å$^2$ total dose each (corresponding to ~16.05 e-/px/s dose rate measured on vacuum). The images were acquired at under-focus optical conditions with a defocus range of −3.3 to −0.9 μm with 0.3 μm steps.

## cryoEM reconstruction

All micrographs (Supplementary Fig. 1a) were inspected and motion-corrected using WARP[47]. After CTF estimation and correction, particles were picked using an automated protocol embedded in WARP. Picked particles were imported to cryoSPARC v3.3.1[48] and the whole reconstruction was performed with this software from this point. Imported particles were submitted to reference-free 2D classification and assigned to 100 classes (Supplementary Fig. 1b) in order to remove false-positive picks. The remaining particles were submitted to subsequent reference-free 2D classifications to select the best possible particles for 3D reconstruction. Ab-initio reconstruction was performed for 5 classes with 100k of randomly selected particles, after which heterogeneous refinement protocol was employed using the complete data set. The most defined 3D class, containing most of the particles, was refined in C1 symmetry using a homogenous refinement protocol. To further clean up the particle set, iterative rounds of 3D classification were used. Particles were subclassified into 5 classes (Supplementary Fig. 1e) without reference volume. The best class, containing ~91% of the remaining particles, were forwarded to the next steps to "homogenous refinement" (Supplementary Fig. 1f) and "local refinement" protocols (Supplementary Fig. 1g) to get the final 3D map. All steps of 3D refinement were done applying C1 symmetry.

## Model fitting, refinement and validation

Structures of monomeric human eIF5A (PDB ID: 3CPF) and tetrameric DHS (PDB ID: 6XXL) were manually docked using UCSF ChimeraX[46] followed by "Dock in Map" tool in Phenix[49]. Further manual model rebuilding was performed in COOT, followed by iterative cycles of real-space refinement in Phenix[50]. Final models were validated using MolProbity. Figures were created in UCSF ChimeraX[46] and PyMOL.

## Thermal stability analysis

To investigate the protein stability and determine its melting temperature we used two complementary methods: Differential Scanning Fluorymetry (DSF) and analysis based on intrinsic fluorescence using a Tycho NT device (Nanotemper, Germany)[51]. For the DSF assay the protein solution (2 mg/ml) was incubated with 1:500 diluted Sypro Orange dye and storage buffer. During measurement fluorescence signal ($\lambda_{ex} = 492$ nm, $\lambda_{em} = 610$ nm) from Sypro Orange was measured as a function of temperature between 5 and 95 °C in increments of 1.2 °C/min. The melting temperature was calculated from the inflexion point of the fluorescence curve. For comparison, protein stability was also assessed using label-free analysis with Tycho NT.6 (Nanotemper).

The capillary was loaded with 10 μL of protein solution (2 mg/ml) and then heated from 35 to 95 °C in 3 min. The melting temperature was calculated from the inflection point of the ratio of 350 nm/330 nm curve. At least three independent repeats were done for each experiment.

## Single turnover fluorescence assay

The single turnover fluorescence assay was performed as described previously[16,33]. Briefly, 15 μM DHS variants were incubated in the presence of 1 mM NAD in 100 mM Tris-Bicine pH 8.5 buffer and the fluorescence excited at 350 nm and fluorescence emission at 441 nm was recorded. After ~2 min, SPD was added to a final concentration of 1 mM and measurement was immediately continued for ~2 min. An observed rapid burst of fluorescence, derived from a rising NADH concentration, was taken as the measure of DHS activity within the first step of its reaction. Each experiment was carried out at least three times.

## FRET measurements

To investigate the DHS affinity to polyamine, a FRET experiment relying on the energy transfer from DHS W327 residue to the di-hydro nicotinamide ring of NADH was performed, as described previously[29]. Briefly, 5 μM of DHS wt was incubated in the presence of 10 μM NAD in 0.2 M Glycine/NaOH pH = 9.3 buffer, and sequentially, 5/10/20 μL of 100 μM SPD were added to the reaction mixture. After the addition of each substrate portion, the fluorescence spectrum ($\lambda_{ex} = 295$ nm, $\lambda_{em} = 441$ nm) was recorded using Shimadzu Fluorescence Spectrometer RF-6000 until saturation was achieved. Experiments were carried out in triplicates. Fluorescence data were normalized for increasing reaction volume and apparent $K_D$ values were calculated using the Hill model implemented in Graphpad Prism 8 (GraphPad Software, Inc., CA, USA). The maximal binding (Bmax value) of DHS wt was used as a reference for final calculations.

## Analysis of the protein oligomeric state by mass photometry

Mass photometry data were collected on a Refeyn OneMP instrument. The instrument was calibrated with BSA standard protein. Ten microliters of selected protein was applied to 10 μL buffer on a coverslip resulting in a final concentration ~1-5 nM. Movies were acquired by using AcquireMP 2.3.0 software followed by data processing in DiscoverMP 2.3.0 software. Masses of DHS variants were estimated by fitting a Gaussian distribution into the mass histograms and taking the value at the apex of the distribution. Probability density functions were extracted and overlaid in GraphPad Prism 8.

## Microscale thermophoresis (MST)

A Monolith NT.115 instrument (NanoTemper Technologies) was used to analyse the complex formation between DHS and eIF5A. To investigate how DHS variants bind to eIF5A, His-tagged DHS proteins were labelled with Monolith His-tag Labeling Kit RED-tris-NTA 2nd Generation (NanoTemper Technologies). Experiments were performed in assay buffer (50 mM Tris, 200 mM NaCl. 5 mM β-ME pH 8). For all the measurements, sixteen eIF5A dilutions in the range of 0.00175–57.5 μM were prepared and then mixed with 8.32 nM of DHS-labelled proteins followed by loading into Monolith NT.115 capillaries. For eIF5A pathological variants, 50 nM labelled His-tagged eIF5A variants were incubated with 0.000122–8 μM DHS wt solutions. Initial fluorescence measurements followed by thermophoresis measurements were carried out using 40% excitation power and 60% MST power, respectively. Data for at least three independent measurements were analysed (MO.Affinity Analysis software, NanoTemper Technologies), allowing for a determination of dissociation constants ($K_D$). The data were presented using GraphPad Prism 8 software. The interactions between DHS variants and eIF5A in the presence of

NAD with or without SPD were carried out in assay buffer supplemented with appropriate ligands to a final concentration of 1 mM.

## Hypusination assay and western blot analysis

The enzymatic activity of DHS variants was assessed by western blot analysis. Five micrograms of His-tagged eIF5A1 was incubated in the presence of 15 μg of appropriate DHS variant in the reaction mixture containing 1 mM NAD, 1 mM SPD in 0.2 M Glycine/NaOH pH = 9.3 with 0.2 M NaCl buffer. Reactions were also performed in the absence of DHS wt and eIF5A as controls. All reaction mixtures were incubated for 1 h at 37 °C and then 100 mM SPD was added to terminate the reaction. Samples were loaded onto an SDS-PAGE gel and blotted to a nitrocellulose membrane (25 mM Tris, 192 mM glycine and 20% methanol) and blocked with 5% skim milk in PBS (pH 7.0) for 1 h at RT. Membranes were incubated for 1 h at RT with a primary rabbit FabHpu98 antibody (Creative Biolabs) diluted 1:4000 in 5% skim milk in PBS (pH 7.0)[52]. Membranes were washed with a T-PBS buffer (PBS supplemented with 0.1% Tween 20). As a next step secondary anti-rabbit-IgG horse-radish, peroxidase-conjugated antibody (Cell signalling) was used (1:2000) for 1 h at RT. The membranes were again washed with T-PBS buffer, followed by development with an enhanced chemiluminescence detection kit (SuperSignal West Pico Plus, Thermo Scientific) according to the manufacturer's instructions.

## Hydrogen/deuterium exchange mass spectrometry (HDX-MS)

Individual eIF5A and DHS$^{K329A}$ proteins and the eIF5A-DHS$^{K329A}$ complex, which was prior HDX-MS constituted and purified by size-exclusion chromatography (see above), were employed at concentrations of 40 μM. Preparation of samples for HDX-MS experiments was aided by a two-arm robotic autosampler (LEAP Technologies) and conducted essentially as described previously with minor modifications[53].

In brief, 7.5 μl of protein solution (eIF5A, DHS$^{K329A}$ or the eIF5A/DHS$^{K329A}$ complex) was mixed with 67.5 μl D$_2$O-containing buffer (25 mM HEPES-Na pH 7.5, 150 mM NaCl, 5 mM β-mercaptoethanol) to initiate the hydrogen/deuterium exchange reaction. After incubation at 25 °C for 10, 30, 95, 1,000 or 10,000 s, 55 μl were withdrawn from the reaction and mixed with 55 μl quench buffer (400 mM KH$_2$PO$_4$/H$_3$PO$_4$, 2 M guanidine-HCl, pH 2.2), which was predispensed and cooled at 1 °C. 95 μl of the resulting mixture was injected into an ACQUITY UPLC M-Class System with HDX Technology[54]. Undeuterated samples were prepared similarly by tenfold dilution in H$_2$O-containing buffer followed by approximately 10 s incubation at 25 °C. The injected quenched HDX reaction was flushed out of the sample loop (50 μl) with constant flow (100 μl/min) of water + 0.1% (v/v) formic acid and guided to a cartridge (2 mm × 2 cm) that was filled immobilized porcine pepsin and digested at 12 °C. The resulting peptic peptides were trapped on a trap column cartridge (2 mm × 2 cm) filled with POROS 20 R2 material (Thermo Scientific) kept at 0.5 °C. After 3 min of trapping, the trap column was placed in line with an ACQUITY UPLC BEH C18 1.7 μm 1.0 × 100 mm column (Waters), and the peptides eluted at 0.5 °C with a gradient of water + 0.1% (v/v) formic acid (A) and acetonitrile + 0.1% (v/v) formic acid (B) at 60 μl/min flow rate as follows: 0−7 min/95−65% A, 7−8 min/65-15% A, 8−10 min/15% A. Eluting peptides were guided to a G2-Si HDMS mass spectrometer with ion mobility separation (Waters), and ionized by electrospray ionization (capillary temperature 250 °C, spray voltage 3.0 kV). Mass spectra were acquired over a range of 50 to 2000 $m/z$ in enhanced high definition MS (HDMSE) or high definition MS (HDMS) mode for undeuterated and deuterated samples, respectively[55]. Lock mass correction was conducted with [Glu1]-Fibrinopeptide B standard (Waters). During chromatographic separation of the peptides, the pepsin column was washed three times with 80 μl of 4% (v/v) acetonitrile and 0.5 M guanidine hydrochloride, and blanks were performed between each sample. Three technical replicates (independent H/D exchange reactions) were measured per incubation time. No correction for HDX back exchange was conducted.

Further data analysis was conducted as described[53]. Peptides were identified with ProteinLynx Global SERVER (PLGS, Waters) from the non-deuterated samples acquired with HDMSE by employing low energy, elevated energy, and intensity thresholds of 300, 100 and 1000 counts, respectively. Hereby, the identified ions were matched to peptides with a database containing the amino acid sequences of eIF5A, DHS$^{K329A}$, porcine pepsin, and their reversed sequences with the following search parameters: peptide tolerance = automatic; fragment tolerance = automatic; min fragment ion matches per peptide = 1; min fragment ion matches per protein = 7; min peptide matches per protein = 3; maximum hits to return = 20; maximum protein mass = 250,000; primary digest reagent = non-specific; missed cleavages = 0; false discovery rate = 100. Deuterium incorporation into peptides was quantified with DynamX 3.0 software (Waters). Only peptides that were identified in all undeuterated samples and with a minimum intensity of 30,000 counts, a maximum length of 30 amino acids, a minimum number of three products with at least 0.1 product per amino acid, a maximum mass error of 25 ppm and retention time tolerance of 0.5 min were considered for analysis. All spectra were manually inspected and, if necessary, peptides omitted (e.g., in case of low signal-to-noise ratio or presence of overlapping peptides).

The observable maximal deuterium uptake of a peptide was calculated by the number of residues minus one (for the N-terminal residue that after proteolytic cleavage quantitatively loses its deuterium label) minus the number of proline residues contained in the peptide (lacking an exchangeable peptide bond amide proton). For the calculation of HDX in per cent the absolute HDX was divided by the theoretical maximal deuterium uptake multiplied by 100. To render the residue-specific HDX differences from overlapping peptides for any given residue of eIF5A or DHS$^{K329A}$, the shortest peptide covering this residue is employed. Where multiple peptides are of the shortest length, the peptide with the residue closest to the peptide's C-terminus is utilized.

An overview of the parameters and characteristics of the HDX-MS experiments is given in Supplementary Table 2.

## Reporting summary

Further information on research design is available in the Nature Portfolio Reporting Summary linked to this article.

## Data availability

The structure coordinates have been deposited in the Protein Data Bank with the PDB IDs 8A0E (eIF5A-DHS), 8A0F (DHS$^{K329A}$), 8A0G (DHS wt with trapped transition state), and 7A6T (DHS N173S in complex with NAD and SPD). The cryo-EM density map for the complex has been deposited in EMDB with the ID: EMD-15052. PDB codes of previously published structures used in this study are 3CPF (monomeric human eIF5A), 6XXL (tetrameric DHS), 5DAT (yeast ribosome in complex with eIF5A), 5DLQ (RanGTP-Exportin 4-eIF5A complex), 6XXJ (wild-type DHS), and 6Q84 (RanGTP-Pdr6-eIF5A export complex). HDX-MS data have been deposited to the ProteomeXchange Consortium via the PRIDE[26] partner repository with the dataset identifier PXD040460. Source data are provided with this paper.

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

## Acknowledgements

The research has been supported by National Science Centre (NCN, Poland) research grant no. UMO-2019/33/B/NZ1/01839 to P.G. and no. 2019/35/N/NZ1/02805 to E.W. X-ray data were collected at the BESSY II 14.1 and 14.3 beamlines at Helmholtz-Zentrum Berlin für Materialien und Energie. We would particularly like to acknowledge the support of Dr. Gert Weber during RT data collection. In addition, we thank the MCB Structural Biology Core Facility (supported by the TEAM TECH CORE FACILITY/2017-4/6 grant from Foundation for Polish Science) for providing instruments and support, in particular, Klaudia Woś for her efforts and assistance during crystallization trials. EW would like to acknowledge the financial support of the DAAD Short Term Grant for her stay at the prof. Bange group. We thank Refeyn Ltd. for the Refeyn OneMP Mass Photometer and especially Tomás de Garay and James Wilkinson for the data acquisition. This publication was developed under the provision of the Polish Ministry of Education and Science project: "Support for research and development with the use of research infrastructure of the National Synchrotron Radiation Centre SOLARIS" under contract nr 1/SOL/2021/2. We acknowledge SOLARIS Centre for the access to the Titan Krios microscope. The 'DFG-core facility for interactions, dynamics and macromolecular assembly structure' at the Philipps-University Marburg supported this work (to G.B.). The open-access publication of this article was funded by the Priority Research Area BioS under the program 'Initiative of Excellence—Research University' at the Jagiellonian University in Krakow.

## Author contributions

P.G. initiated, conceived, and supervised the study. E.W. and P.G. designed the experiments. E.W. performed molecular cloning, protein expression, purification, and all functional assays. E.W. and P.W. crystallized protein, performed diffraction data collection, solved, and refined crystal structures; M.R. performed cryo-EM sample preparation, data collection, and analysis, A.B. performed cryo-EM data analysis and reconstructions; E.W. and P.W. performed cryo-EM model building and refinement; E.W. and W.S. performed HDX-MS experiments; E.W., P.W., A.B., K.M.Z., W.S., G.B., S.G., and P.G. analyzed the data. E.W., P.W., W.S. and A.B. prepared figures. P.W. prepared the movie. E.W. and P.G. wrote the manuscript with assistance from the other authors.

## Competing interests

The authors declare no competing interests.
