## [Peer Review File · Nature Communications]

Cryo-EM structure of human eIF5A-DHS complex reveals the molecular basis of hypusination-associated neurodegenerative disordersREVIEWER COMMENTS

Reviewer #1 (Remarks to the Author):

Dr. Grudnik and colleagues discuss the molecular details of the DHS-mediated hypusination of eIF5A. The study is significant as it will better our understanding of how clinically-relevant mutations of DHS and eIF5A affect this cellular process leading to several neurological disorders.

My questions and suggestions for the authors are as follows:

Results:

Line 82: Please check the reference for the previously reported crystal structures. It is cited as ref 1, but the paper titled "Identification of the hypusine-containing protein Hy+ as translation initiation factor eIF-4D " does not discuss the mentioned crystal structures.

Line 83: 'chain A of the loop' should be 'the loop of chain A'

Line 143: Can be separated into a separate paragraph to differentiate the results of eIF5a and DHS compared to complex

Line 150-151: what do the authors mean by the heterogeneity of eIF5A-bound and empty DHS molecules? Is it because the Cryo-EM structure didn't show any difference between the bound and unbound states? How does the mixture exist if the authors used size exclusion chromatography fractions corresponding to the complex?

I suggest the authors refrain from referring to regions of bimodal distribution as "areas of increased or decreased H/DX."

Add peptides shown in Extended Fig. 3, 4, and 5 in parentheses in text and figure legends.

Please mention back exchange %

Methods:

In the H/DX experiment section, please mention if the first two amide hydrogen were ignored for the calculation of deuterium incorporation.

Figures:

In Extended Data Fig. 3a and Extended Data Fig. 4a, please depict residues corresponding to the binding site/active site above the sequence.

Extended Data Fig. 3b and Extended Data Fig. 4b: Replace the heatmap showing delta D (Complex - protein) vs. individual residue with a heatmap showing peptides vs. delta D. The heatmap can cover the peptides from regions of representative curves shown in Extended Data Fig. 3c.

In Extended Data Fig. 3c, Extended Data Fig. 4c, and Extended Data Fig. 5e,f the font is too small and difficult to read.

In Extended Data Fig. 3d and Extended figure 4d, the exchange mapping at each time point can be removed and can replace the combined exchange figure with Extended Data Fig. 5a and c (which highlights only peptides showing EX2 kinetics)

In Extended Data Fig. 5b, the authors did not show the region 24-30. However, the text mentions

this as an area of bimodality.

In Extended Data Fig. 5e and f, the kinetic curve is not a correct representation of the mixed population (structures and less structured). Therefore, I suggest the authors remove the plots above the bimodal distribution spectra.

In Extended Data Fig. 5e, from the Isotopic distribution of peptide (32-42), the spectra for no HDX of both eIF5A and complex look like an overlap of 2 peptides exists.

In Extended figure 3c, the authors provided the curve for peptide spanning residues 34-40; in Extended figure 5e, peptide spanning residues 32-42 is shown. Was the peptide 34-40 showing bimodal distribution?

Please provide spectra for representative peptides showing EX2 kinetics.

Reviewer #2 (Remarks to the Author):

The manuscript by Wator et al. addresses the structural and biochemical description of hypusination mechanism through eIF5A-DHS complex. More specifically, the authors use mainly single particle cryo-EM, protein crystallography and biochemistry to study deoxyhypusine synthase (DHS) in free and eIF5 bound form. This work sheds more light into molecular details of the hypusination reaction and with mutagenetic studies reveal the roles of individual residues required for this reaction. The cryo-EM structure together with hydrogen-deuterium exchange experiments reveal the dynamic behaviour of the complex. The authors provide extensive biochemical and mutagenesis evidence of clinically relevant mutations which are associated with neurodegenerative disorders. Based on the cryo-EM structure together with X-ray structures (some representing the intermediate state) the authors claim to uncover the details of the hypusination reaction.

Overall, this work is presented well. The strong side of this work is the combination of a structural approach with biochemical techniques. In summary, the results of this work could, in principle, be interesting to a broad readership of Nature Communications. However, there are several major concerns denoted below that need to be addressed by the authors, before considering the manuscript for publication in Nature Communications.

Major issues:

1. The authors present the refinement statistics for X-ray structures of the DHS in many forms. However, this reviewer considers the statistics to be highly overestimated for all three structures, especially in the highest resolution shell. Even though the completeness is high, the I over sigma values together with CC(1/2) values are showing high contribution of the background in the collected datasets. Especially looking at I over sigma values the background has an equal or even twice that high signal than the real protein signal. The reviewer strongly suggests to revisit the datasets and cut the data at a lower resolution shell to significantly improve the statistics (as the resolutions presented in the paper by the authors do not represent the true value).
2. The manuscript presents also a cryo-EM structure of the eIF5A-DHS complex. However, the authors in the table show that the symmetry imposed was O. This usually represent a 5 fold symmetry axis. Based on the particle itself, the particle has only the symmetry of identity C1. This reviewer is therefore very hesitant to say if the map was correctly reconstructed or if this is only a typo in the table. If the reconstruction was really performed in O symmetry the whole map is misrepresented.
3. The authors did not place the details of the processing in the Methods section. This has to be included. The authors need to address all the packages used for 2D classification and 3D classification. The strategy is highlighted in the Extended Data Fig. 1, but this is not sufficient. A section in Methods has to be devoted to cryo-EM data processing and structure refinement. The most crucial part is also the structure refinement procedure into the cryo-EM map, which is

absolutely missing. The authors have to provide information how was the structure refined. Again to report a table with validation statistics is not sufficient. Moreover, the table is missing some very important information, like pixel size, final images used for reconstruction.

Minor issues:

4. The reviewer recommends that the authors would check the manuscript once again for some typos or additional words, which are not necessary in the sentences.
5. Line 99-100, The authors should clarify which chain of DHS (either A or B) with the outermost helix flanks the binding site of eIF5A.
6. Line 223-224, The authors describe conformation freedom for Trp327, in Figure 2f they highlight two conformations. Can they elaborate on the percentage what conformation is favorable more?
7. Fig 1c, from the perspective of the panel, it seems like W327 is in clash with the main chain, can the authors use a slightly rotated view?
8. Fig 1d, the authors use AS and BnC abbreviations, can they explain them in the figure legend?
9. Fig 3a, the authors show the melting temperatures for mutated DHS. However the only mutant with significant difference is for the mutant I271A. The authors should revise their text accordingly when describing the stability of DHS mutants, as this one exceeded 2 degrees of Celsius difference.
10. Extended data Fig. 2 b and d, please add labels for NAD, lysine or modified version. Also highlight the terminal ends of eIF5A.
11. Extended data Fig 6b, please use less thicker mesh for electron density to make the K*329 more visible.

Reviewer #3 (Remarks to the Author):

To understand the molecular mechanism how deoxyhypusine synthase (DHS) binds eukaryotic initiation factor is of importance for the scientific community working in drug discovery on the hypusine pathway. Dr. Grudnik who coauthors this manuscript, has applied a structural biology approach to answer this question. In the translational part of the manuscript the authors investigated DHS-eIF5A complex formation in a human DHS variant using the same structural biology methods as they did for the crystallized, human wild protein. They discovered that complex formation between both proteins, i.e. DHS and EIF5A was hampered. The manuscript intensifies and completes previous studies on crystallized human DHS protein by Dr. Park who performed the fundamental work to crystallize the human DHS protein (Structure 6, 23-32 (1998)). Dr. Grudnik confirmed and clarified within the experimental part that the ball-and-chain motif is critical for the binding of the substrates i.e., eIF5A and spermidine. The most important novel findings presented in this manuscript comprise the DHS variant DHSN173S and the pathological deletion variant DHSdel305Y_306I variant. While DHS variant DHSN173S is hampered in binding of spermidine but still forms deoxyhypusine, the deletion variant is a loss of function mutant which can not bind the substrate spermidine anymore.

The experimental data are convincing and sound. The analytical methods are justified to tackle the scientific questions. What I am missing is the quantitative determination of DHS specific activities in case of the DHS variant DHSN173S and the loss of function mutant DHSdel305Y_306I. Moreover, as a proof of principle, quantified, specific DHS enzyme activities of the mutant constructs showing the amino acids involved in the interface between EIF5A and the DHS protein (Fig.3C) should be determined. So far Western Blot experiments (hypusination capacity) were performed and relative oxidoreductase activities were determined (Fig.3B) expressed in %? In this

context, it is of interest to mention that a monoclonal antihypusine antibody was applied according to citation 45.

The quality and presentation of the data is appropriate. Hence, I have some suggestions: The graphs in Figure 3c and d are difficult to follow. Different coloured circles are presented. Instead, besides circles, triangles, squares should be applied. I am missing an explanation for BnC in the legend of Fig.1d. Some typos appear in the text: "apo enzyme", "conduced" etc. Why is citation 14 in italic?

Conclusions of this manuscript are in accordance with the data presented. The significance of this manuscript i.e., the elucidation of interaction of DHS with its substrate eIF5A will support the discovery of small molecules against the human DHS protein. Hitherto, only spermidine mimetics have been applied which are not target specific.

Reviewer #4 (Remarks to the Author):

In this manuscript, Wator and colleagues aimed to discover the molecular details of the deoxyhypusination process and to provide insights into the effects of deoxyhypusine synthase (DHS) mutations. To this end, they elucidated the cryo-EM structure of eIF5A in complex with DHS. The model illustrates how eIF5A, specifically the hypusine-containing loop, is recognized at the active site. The authors later used revealed interactions to generate DHS mutants and characterized them for stability and activity of DHS. Furthermore, they investigated clinically-relevant DHS mutations and disclosed their impact on eIF5A interaction. Remarkably, the authors also captured DHS in a reaction intermediate and presented its X-ray crystal structure. Using both structures they proposed how deoxyhypusination reaction takes place at the molecular level. The manuscript is exciting and provides valuable insights into the deoxyhypusination process, one of the ancient and essential reactions. The experiments use state of the art techniques. The data is verified by complementary methods and, in general, supports the conclusion drawn. Several points should be addressed before publication.

1) The manuscript will greatly benefit from having a close-up view of eIF5A (and hypusine/ lysine loop) and the surrounding active site interactions. Figures 1c and 2a (even Fig. 2c) have binding site views but each presents side chains of a single component. The supplementary video beautifully depicts such a recognition. However, given that the eIF5A recognition is one of the most important results of the study, presenting such an interface directly as a main display item will add strength to the manuscript.

2) Accordingly, Figure 2a should be moved to Figure 3 with a similar orientation as in Fig. 3f-h. Otherwise, readers will need to scroll back and forth to locate where the mentioned mutations (mentioned in Fig.3) are.

3) The cryo-EM structure shows a single eIF5A bound to tetrameric DHS, which looks in line with previous findings. Given the similarities and differences to DHS apo structures, is there any structural explanation why a single eIF5A (out of 4 binding sites) is present? Could this be influenced by the sample preparation? For example, was there a stoichiometric or excess eIF5A during complex formation? In any case, sample preparation should be described in the methods section.

4) Figure 3d and 3e need revision. At the moment, color coding and labelling in legends are inconsistent. Similarly, coloring of the curves look mixed up. For instance, in Fig. 3d, there is no green curve corresponding to D243A. The authors should make sure similar errors are not present in 3e. In addition, color choice for the curves is not ideal. Similar shades were often used for curves that are far apart. The authors should reconsider their palette choice or simplify the coloring. Alternatively, curves from left to right could be in the same order as legends top to bottom or so.

5) In HDX-MS experiments, changes in deuterium exchange rates could hint conformational alterations or they could simply indicate increased or decreased stability upon binding of the interacting residues. How can one separate which one is the predominant one? How can the

authors be so sure that this leads to "folding/unfolding of secondary structures" (Line 140-141) in the context of eIF5A-DHS interaction. They suggest for example that C-terminal of eIF5A undergoes conformational changes upon binding to DHS. This region does not interact with DHS, so can they comment on how and why such a folding/unfolding occurs? In general, findings of this section will benefit from further discussion.

6) In Extended Data Fig. 5e and f, the font size is too small to read. Accordingly, it is hard to distinguish the colors and corresponding structure segments in panels a, b and d.

7) Extended Data Fig. 7 should be renumbered as Data Fig. 6 to keep the logical order in the text.

8) In line 190-192, "Mutations in active site residues (H288A and K329A), had no or only minor effects on eIF5A binding (K_D 49.7 ± 28.7 nM and 128.6 ± 37.9 nM)", reduction of K329A mutant's affinity to 128 nM is considered as "only minor". While in line 316-318, similar reduction in N173S and deletion mutants were considered significant and led to the further conclusions. One of these statements require revision.

9) Quality of crystallographic data look very nice. Nevertheless, RSRZ outliers seem to be an issue in all three X-ray structures. So, I wonder if these point to a general problem in the refinement strategy. Details of the validation reports reflect significant number of outliers in fit of the model and data, and even include some of the ligands. I don't believe that they will affect the conclusions of the text, yet I invite authors to investigate potential problems either in the data or in the refinement approach.

10) Crystallographic table: all numbers should consistently have 1 or 2 decimal digits.

11) In electrostatic potential images, blue and red generally refer to max and min, respectively, since they are also used to depict nitrogen and oxygen atoms. Using inverse coloring in Figure 1d could mislead readers, although it is explicitly explained in the color key and also in the legends.

Typos/Grammar

12) Line 97 Likewise (no However)

13) Line 148 homo-tetramer (no underscore)

14) Supplementary Line 47 (a, c) Differences in HDX of eIF5A (not a and b)
Line 49 (b, d) areas of eIF5A... (not c and d)

REVIEWER COMMENTS

Reviewer #1 (Remarks to the Author):

Dr. Grudnik and colleagues discuss the molecular details of the DHS-mediated hypusination of eIF5A. The study is significant as it will better our understanding of how clinically-relevant mutations of DHS and eIF5A affect this cellular process leading to several neurological disorders.

My questions and suggestions for the authors are as follows:

Results:

Line 82: Please check the reference for the previously reported crystal structures. It is cited as ref 1, but the paper titled “Identification of the hypusine-containing protein Hy⁺ as translation initiation factor eIF-4D ” does not discuss the mentioned crystal structures.

Response: The inappropriate reference was removed from the statement. References 16, 18 and 23 correctly refer to previously reported crystal structures.

Line 83: ‘chain A of the loop’ should be ‘the loop of chain A’

Response: The sentence was corrected.

Line 143: Can be separated into a separate paragraph to differentiate the results of eIF5a and DHS compared to complex

Response: Modified according to the reviewer’s suggestion

Line 150-151: what do the authors mean by the heterogeneity of eIF5A-bound and empty DHS molecules? Is it because the Cryo-EM structure didn’t show any difference between the bound and unbound states? How does the mixture exist if the authors used size exclusion chromatography fractions corresponding to the complex?

Response: In the cryo-EM model, we observe the binding of eIF5A in only one of four DHS active sites. This single binding event does not induce any major conformational change in DHS. However, we measured Hydrogen-Deuterium Exchange rates and found that the same peptide might exhibit different states of solvent exposure in the same sample, indicating bimodality. In this sense, heterogeneity refers to small conformational differences between individual peptides originating from the occupied and unoccupied active sites – those changes are not detectable by size exclusion chromatography.

The mixture of DHS and eIF5A-DHS exists due to the relatively small size difference between the protein alone (164 kDa) and the formed complex (181 kDa). Even after applying the reconstituted complex onto a size exclusion chromatography column, we were not able to obtain clearly separated sample (see Review Figure 1). To enrich the reconstituted complex for our cryo-EM and HDX analyses, we only used the shoulder of the elution peak (Rev. Fig. 1, fraction no. 2, marked green). From the SDS-page analyses, we can judge that the fraction from the main part of the peak contained different proteins ratio. Of note, we could not achieve stoichiometric saturation of DHS by eIF5A in our size-exclusion chromatography analyses, despite using a molar excess of eIF5A and different protein variants.

Review Figure 1. Reconstitution of the eIF5A-DHS complex. SEC profile of the eIF5A-DHS K329A complex (purple) and DHS K329A (grey). For cryoEM grid preparation, fraction no. 2 (green) was used. SDS-PAGE of the individual complex components and fractions.

Foremost, we apologize for not including the description of the complex reconstitution in the original version of the manuscript. We have now added the following paragraphs in the Method section of the revised version of the manuscript:

“Complex reconstitution

We reconstituted the human eIF5A-DHS^{K329A} complex using previously purified individual components, namely eIF5A-1 and catalytically inactive DHS^{K329A}. DHS^{K329A} was mixed with a molar excess of eIF5A-1 (2 mole eIF5A per 1 mole DHS). Proteins were incubated in 200 mM Glycine/NaOH pH=9.3 buffer with 200 mM NaCl in the presence of 1 mM NAD and 1 mM SPD. The complex was then purified by size-exclusion chromatogram using a Superdex200 Increase 10/300 column (GE Healthcare) pre-equilibrated with 200 mM Glycine/NaOH pH=9.3 buffer with 200 mM NaCl. Due to the very small difference in mass of the complex relative to DHS alone the fraction from the very front of the elution peak was used for further investigation.”

I suggest the authors refrain from referring to regions of bimodal distribution as “areas of increased or decreased H/DX.”

Response: We amended the description and presentation of the HDX-MS results. We do now put more emphasis on the proper designation of bimodality in both, figures and text.

The explicit mentioning of the weakness of HDX in this particular case (the 1:4 stoichiometry of the eIF5A-DHS complex) to discern eIF5A-bound and unbound DHS molecules should safeguard the reader from the premature interpretation of our data. Along this line (and the subsequent concern of this reviewer), we called up the peptides and residues properly in this section to draw the connection between the difference in total observed HDX and the presence of bimodal distributions.

Add peptides shown in Extended Fig. 3, 4, and 5 in parentheses in text and figure legends.

Response: We did not mention all of the displayed peptides in the text but mentioned a few selected ones – now also referring to the respective figures displaying all peptides that are not explicitly mentioned in the text. We hope that this is an acceptable solution for the reviewer.

Please mention back exchange %

Response: As the HDX-MS experiment directly compared the behavior of eIF5A, DHS and the eIF5A-DHS complex, and we only provided further evidence on top of the cryo-EM structure for the formation of the complex, the determination of back exchange was not a critical requirement of this particular analysis. Hence, no determination of back exchange was performed. This is now specifically denoted in the materials and methods section.

Methods:

In the H/DX experiment section, please mention if the first two amide hydrogen were ignored for the calculation of deuterium incorporation.

Response: Only the first residue was ignored for the calculation of deuterium calculation. We amended the materials and methods section, which now reads as follows:

“The observable maximal deuterium uptake of a peptide was calculated by the number of residues minus one (for the N-terminal residue that after proteolytic cleavage quantitatively loses its deuterium label) minus the number of proline residues contained in the peptide (lacking an exchangeable peptide bond amide proton).”

Figures:

In Extended Data Fig. 3a and Extended Data Fig. 4a, please depict residues corresponding to the binding site/active site above the sequence.

Response: We now show the secondary structure elements above the amino acid sequences in Extended Data Figs. 3a and 4a. Furthermore, we highlight binding and active site residues in DHS, and Lys50 in the hypusination loop of eIF5A.

Extended Data Fig. 3b and Extended Data Fig. 4b: Replace the heatmap showing delta D (Complex - protein) vs. individual residue with a heatmap showing peptides vs. delta D. The heatmap can cover the peptides from regions of representative curves shown in Extended Data Fig. 3c.

Response: We expanded the depiction of peptides identified in the HDX experiment (formerly only as black bars) in Extended Data Figs. 3a and 4a by displaying the difference in HDX for these peptides, respectively. However, we prefer keeping the difference in HDX per residue (Extended Data Figs. 3b and 4b) in the figures, as we felt this plot would be necessary to assert the mappings of HDX data onto our structural models.

In Extended Data Fig. 3c, Extended Data Fig. 4c, and Extended Data Fig. 5e,f the font is too small and difficult to read.

Response: We agree that in the previous version of the figures, the font size was too small. We now increased the font size in all Extended Data Figures presenting data gathered by HDX-MS.

In Extended Data Fig. 3d and Extended figure 4d, the exchange mapping at each time point can be removed and can replace the combined exchange figure with Extended Data Fig. 5a and c (which highlights only peptides showing EX2 kinetics).

Response: We believe that the projection of the obtained HDX data on the structures right beneath the mapping on the amino acid sequences is an acceptable way of representing these data. This representation enables the reader to directly grasp where changes are located in the structure.

In Extended Data Fig. 5b, the authors did not show the region 24-30. However, the text mentions this as an area of bimodality.

Response: The region of eIF5A spanning residues 24-30 exhibited higher deuterium incorporation in the complex than in the individual protein but no apparent bimodality. We heavily amended the results section describing the HDX-MS data and specifically show the areas where differences in HDX (of the entire population) appear. In addition, we now highlight the areas, which are linked to the appearance of mixed populations/bimodal behavior of HDX.

In Extended Data Fig. 5e and f, the kinetic curve is not a correct representation of the mixed population (structures and less structured). Therefore, I suggest the authors remove the plots above the bimodal distribution spectra.

Response: We agree that the side-by-side representation of deuterium uptake curves and mass spectra is not the technically most appropriate way of showing these data. However, Deuterium uptake curves are easily understandable by a broad readership and we would prefer to keep them. Thus, deuterium uptake curves now appear in a separate figure in which we marked those peptides clearly showing mixed populations with black boxes. Two dedicated extended data figures now display the mass spectra of those highlighted peptides, for which mixed populations are apparent, and other peptides for which this apparently cannot be observed.

In Extended Data Fig. 5e, from the Isotopic distribution of peptide (32-42), the spectra for no HDX of both eIF5A and complex look like an overlap of 2 peptides exists.

Response: We agree that the isotopic distribution of peptide 32-42 indicates the presence of an overlapping peptide. However, the reinspection of the isotopes did not substantiate this notion, as there were neither major deviations from the theoretical masses for any of the isotopes nor differences in ion mobility. We attribute the unusual pattern to a saturation of the detector by this rather abundant peptide.

In Extended figure 3c, the authors provided the curve for peptide spanning residues 34-40; in Extended figure 5e, peptide spanning residues 32-42 is shown. Was the peptide 34-40 showing bimodal distribution?

Response: We apologize for this inconsistency between the figures. Indeed, for the eIF5A peptide spanning residues 34-40 bimodality is not immediately apparent. We now provide the spectra for both peptides.

Please provide spectra for representative peptides showing EX2 kinetics.

Response: According to the reviewer's suggestion, we now provide the spectra of further peptides with pure EX2 kinetics.

Reviewer #2 (Remarks to the Author):

The manuscript by Wator et al. addresses the structural and biochemical description of hypusination mechanism through eIF5A-DHS complex. More specifically, the authors use mainly single particle cryo-EM, protein crystallography and biochemistry to study deoxyhypusine synthase (DHS) in free and eIF5 bound form. This work sheds more light into molecular details of the hypusination reaction and with mutagenetic studies reveal the roles of individual residues required for this reaction. The cryo-EM structure together with hydrogen-deuterium exchange experiments reveal the dynamic behaviour of the complex. The authors provide extensive biochemical and mutagenesis evidence of clinically relevant mutations which are associated with neurodegenerative disorders. Based on the cryo-EM structure together with X-ray structures (some representing the intermediate state) the authors claim to uncover the details of the hypusination reaction.

Overall, this work is presented well. The strong side of this work is the combination of a structural approach with biochemical techniques. In summary, the results of this work could, in principle, be interesting to a broad readership of Nature Communications. However, there are several major concerns denoted below that need to be addressed by the authors, before considering the manuscript for publication in Nature Communications.

Major issues:

1. The authors present the refinement statistics for X-ray structures of the DHS in many forms. However, this reviewer considers the statistics to be highly overestimated for all three structures, especially in the highest resolution shell. Even though the completeness is high, the I over sigma values together with CC(1/2) values are showing high contribution of the background in the collected datasets. Especially looking at I over sigma values the background has an equal or even twice that high signal than the real protein signal. The reviewer strongly suggests to revisit the datasets and cut the data at a lower resolution shell to significantly improve the statistics (as the resolutions presented in the paper by the authors do not represent the true value).

Response: We thank the reviewer for all the valuable comments. We understand and appreciate the reviewer's concerns regarding the quality of the crystallographic data. The criteria for selecting and trimming the resolution of crystal structures have evolved over the past years with the advancement of both software and instrumentation (e.g. the advent of pixel detectors). Along with these changes, the quality criteria of the structures have been adjusted. All of this together has led to a situation, where the resolutions reported in the PDB entry differ in the resolution cut-off assumptions made. To provide additional guidance for the readers, we have updated our crystallographic data table and now also provided resolution values for the cut-off at $I/\sigma I = 2$ in the footnote of the table. Nevertheless, we prefer to keep the presented structures and we would like to explain our reasoning.

The aim of the data processing is to include as much experimental information as possible and to avoid noisy or bad data. Therefore, the cut in resolution is important. The discussion regarding data inclusion or rejection was vibrant in the last two decades and the community agrees that correlation coefficients ($CC_{1/2}$) rather than Rmerge, Rsym, $I/\sigma I$ or similar criteria alone should be applied.

See for example:

- Linking crystallographic model and data quality. P.A. Karplus & K. Diederichs (2012) Science 336:1030-3

- Assessing and maximizing data quality in macromolecular crystallography. P.A. Karplus & K. Diederichs (2015) *Cur. Op. in Str. Biology* 34:60-68
- Better models by discarding data? P.A. Karplus & K. Diederichs (2013) *Acta Cryst. D*59:1215-1222

A sophisticated implementation of resolution cutoff criteria is implemented in the newest version of XDSAPP, which we used for data reduction. In each case, the statistical analyses indicated, that the outermost resolution shell carries significant information. Moreover, the usage of contemporary refinement software based on the maximum likelihood approach allows the inclusion of weak signals thanks to optimal X-ray term weighting.

During structure refinement, both experimental data (X-ray term) and prior knowledge of the ideal chemical environment (model) are taken into account and weighted depending on several parameters. We compared several refinement protocols and used the weight optimization protocols built-in in the phenix.refine software. We believe that despite the presence of an individual violation of ideal geometry, particularly in the low-density regions our structures optimally balance experimental and theoretical data and present the best obtainable models.

Ultimately the data are considered useful and should be included in the analyses if it can improve the model. Paired refinement is currently the most optimal protocol used for data resolution cutoff, in which a model is refined against a particular dataset truncated at various resolution limits and the results are compared using the same set of reflections. A given resolution shell is accepted only if its inclusion improved the model.

Following the comment of the reviewer, we used paired refinement protocol implemented in PAIREF software (Maly et al. *IUCrJ* 2020; Maly et al. *Acta Cryst F* 2021) where our crystallographic models were automatically refined against five incremental resolution limits (0.1 Å step starting from resolution 0.5 Å lower than our final resolution). In all cases, it proved the inclusion of all reflections up to the originally applied cutoff to be beneficial for model refinement.

Notably, the lowest correlation coefficient in our crystal structures is above 0.45, whilst in cryo-EM, the "golden standard" refers to FSC=0.147. Last but not least the resolution is only one of the quality parameters and our primary concern is the fit of the model to experimental data.

Review Table 1. Statistics for resolution cut-off at $I/\sigma I=2$

	7A6T	8A0F	8A0G
I/ σI	2.02	1.97	2.01
Resolution [Å]	1.91	1.84	1.99
CC1/2 [%]	87.6	83.3	79.2
ISa	20.95	20.73	15.63

2. The manuscript presents also a cryo-EM structure of the eIF5A-DHS complex. However, the authors in the table show that the symmetry imposed was O. This usually represent a 5 fold symmetry axis. Based on the particle itself, the particle has only the symmetry of identity C1. This reviewer is therefore very hesitant to say if the map was correctly reconstructed or if this is only a typo in the table. If the reconstruction was really performed in O symmetry the whole map is misrepresented.

Response: We thank the reviewer for pointing us this typo and we apologize for this oversight. No symmetry operator was enforced and C1 was used during the reconstruction (as stated here by the reviewer). The Extended Data Table 1. was corrected and updated accordingly.

3. The authors did not place the details of the processing in the Methods section. This has to be included. The authors need to address all the packages used for 2D classification and 3D classification. The strategy is highlighted in the Extended Data Fig. 1, but this is not sufficient. A section in Methods has to be devoted to cryo-EM data processing and structure refinement. The most crucial part is also the structure refinement procedure into the cryo-EM map, which is absolutely missing. The authors have to provide information how was the structure refined.

Response: We acknowledge and appreciate the reviewer's suggestion here. We apologize for not including a detailed description of cryo-EM reconstruction, model building and refinement procedures in the initial version of the manuscript. We have now added the following paragraphs in the Method section:

cryo-EM reconstruction:

All micrographs were inspected and (Extended Data Fig.1a) motion-corrected using WARP⁴⁷. After CTF estimation and correction, particles were picked using an automated protocol embedded in WARP. Picked particles were imported into cryoSPARC v3⁴⁸ and the whole reconstruction was performed with this software from this point. Imported particles were submitted to reference-free 2D classification and assigned to 100 classes (Extended Data Fig. 1b) in order to remove false-positive picks. The remaining selected particles were submitted to subsequent reference-free 2D classifications to select the best possible particle set for 3D reconstruction. Ab-initio reconstruction was performed for 5 classes with 100 000 of randomly selected particles, after which a "heterogeneous refinement" protocol was employed using the complete data set. The most defined 3D class, containing most of the particles, was refined in C1 symmetry using a homogenous refinement protocol. To further clean up the particle set, iterative rounds of 3D classification were used. Particles were subclassified into 5 classes (Extended Data Fig. 1e) without reference volume. The best class, containing ~91% of the remaining particles, were forwarded to the next steps to "homogenous refinement" (Extended Data Fig. 1f) and "local refinement" protocols (Extended Data Fig. 1g) to get the final 3D map. All steps of 3D refinement were done applying C1 symmetry.

Model fitting, refinement and validation

Structures of monomeric human eIF5A (PDB: 3CPF) and tetrameric DHS (6XXL) were manually placed using UCSF ChimeraX⁴⁶ followed by docking using the "Dock in Map" tool in Phenix⁴⁹. Further manual model rebuilding was performed in COOT, followed by iterative cycles of real-space refinement in Phenix⁵⁰. Final models were validated using MolProbity. Figures were created in UCSF ChimeraX⁴⁶ and PyMOL.

Again to report a table with validation statistics is not sufficient. Moreover, the table is missing some very important information, like pixel size, final images used for reconstruction.

Response: We apologize for this oversight. We have now added pixel size and the number of particles used for the map reconstruction to the Extended Data Table 1.

Minor issues:

4. The reviewer recommends that the authors would check the manuscript once again for some typos or additional words, which are not necessary in the sentences.

Response: We have re-examined the whole manuscript for typos and additional words. Furthermore, we have consulted the revised text with a native speaker to further improve the language.

5. Line 99-100, The authors should clarify which chain of DHS (either A or B) with the outermost helix flanks the binding site of eIF5A.

Response: The detected eIF5A molecule is bound adjacent to subunit B. However helices 173-196 from both subunits symmetrically flank the entrance to the active site and undergo rearrangement of the same magnitude. We have now specified this in the revised text (line 100), which reads as follows – *“Furthermore, we measured a 7° difference in the relative orientation of the DHS outermost helix (chain B) that flanks the binding site of eIF5A.”*

6. Line 223-224, The authors describe conformation freedom for Trp327, in Figure 2f they highlight two conformations. Can they elaborate on the percentage what conformation is favorable more?

Response: Both observed conformations of Trp327 in the crystal structure of DHS^{K329A} are geometrically feasible and are distributed equally with refined occupancies 0.47 vs. 0.53 in subunit A and 0.52 vs. 0.48 in subunit B (with the later value corresponding to the wild-type like conformation). We have also modified the sentence describing the two alternative conformations in the text (lines 311-314):

“In the structure of DHS^{K329A}-NAD-SPD, we observed two alternative conformations of the Trp327 side chain with almost equivalent occupancies. In detail, we found the perpendicular position that is also found in the eIF5A-DHS complex and the position that is parallel to the bound SPD molecule, as in DHS^{wt} (without protein ligand) (Fig. 4c).”

7. Fig 1c, from the perspective of the panel, it seems like W327 is in clash with the main chain, can the authors use a slightly rotated view?

Response: We agree that in the previous version of the Fig. 1c W327 seemed to clash with the main chain. In the updated version of the manuscript, we used two views differing by 30° to show the detailed architecture of the active site.

8. Fig 1d, the authors use AS and BnC abbreviations, can they explain them in the figure legend?

Response: We added the explanations of the abbreviations to the legend of Fig. 1.

9. Fig 3a, the authors show the melting temperatures for mutated DHS. However the only mutant with significant difference is for the mutant I271A. The authors should revise their text accordingly when describing the stability of DHS mutants, as this one exceeded 2 degrees of Celsius difference.

Response: We amended the text accordingly, which now reads as follows –

“Most of the mutations analyzed do not or only slightly affect the stability of DHS, but the mutation of hydrophobic interface residue (I271A) showed significantly decreased melting temperature (Fig. 3a).”

10. Extended data Fig. 2 b and d, please add labels for NAD, lysine or modified version. Also highlight the terminal ends of eIF5A.

Response: We updated Extended Data Fig. 2b and 2e (old 2d) according to the reviewer’s suggestion.

11. Extended data Fig 6b, please use less thicker mesh for electron density to make the K*329 more visible.

Response: We updated the Extended Data Fig. 9b (previously Extended Data Fig. 6b) with less thicker mesh according to the suggestion.

Reviewer #3 (Remarks to the Author):

To understand the molecular mechanism how deoxyhypusine synthase (DHS) binds eukaryotic initiation factor is of importance for the scientific community working in drug discovery on the hypusine pathway. Dr. Grudnik who coauthors this manuscript, has applied a structural biology approach to answer this question. In the translational part of the manuscript the authors investigated DHS-eIF5A complex formation in a human DHS variant using the same structural biology methods as they did for the crystallized, human wild protein. They discovered that complex formation between both proteins, i.e. DHS and EIF5A was hampered. The manuscript intensifies and completes previous studies on crystallized human DHS protein by Dr. Park who performed the fundamental work to crystallize the human DHS protein (Structure 6, 23–32 (1998)). Dr. Grudnik confirmed and clarified within the experimental part that the ball-and-chain motif is critical for the binding of the substrates i.e., eIF5A and spermidine. The most important novel findings presented in this manuscript comprise the DHS variant DHSN173S and the pathological deletion variant DHSdel305Y_306I variant. While DHS variant DHSN173S is hampered in binding of spermidine but still forms deoxyhypusine, the deletion variant is a loss of function mutant which can not bind the substrate spermidine anymore. The experimental data are convincing and sound. The analytical methods are justified to tackle the scientific questions.

What I am missing is the quantitative determination of DHS specific activities in case of the DHS variant DHSN173S and the loss of function mutant DHSdel305Y_306I. Moreover, as a proof of principle, quantified, specific DHS enzyme activities of the mutant constructs showing the amino acids involved in the interface between EIF5A and the DHS protein (Fig.3C) should be determined. So far Western Blot experiments (hypusination capacity) were performed and relative oxidoreductase activities were determined (Fig.3B) expressed in %? In this context, it is of interest to mention that a monoclonal antihypusine antibody was applied according to citation 45.

Response: We thank the reviewer for the valuable comments, suggestions and corrections. Indeed we have previously missed to include a proper reference for the hypusine-specific antibody - this has been now amended and the reference appeared both, in the methods section and the figure legend. We also agree that the presentation of DHS activities could have been confusing. As stated in the Methods section we use an oxidoreductase single-turnover fluorescence assay to assess the DHS activity - as described previously by *Afanador et al* (Structure 2018). In this assay, we compare values of maximal fluorescence burst associated with NAD reduction. Hence we present the activity of the DHS variants in the percentage of the wild-type activity. This confusion has now been amended in the figures. Moreover, we have added a new panel, showing the comparison of specific activities of DHS^{N173S} and DHS^{del305Y_306I} variants (Extended Data Fig. 8d).

The quality and presentation of the data is appropriate. Hence, I have some suggestions: The graphs in Figure 3c and d are difficult to follow. Different coloured circles are presented. Instead, besides circles, triangles, squares should be applied.

Response: We agree that in the previous version of the manuscript the graphs were difficult to follow. We have now made new graphs to replace previous panels d and e.

I am missing an explanation for BnC in the legend of Fig.1d.

Response: The figure legend has been amended. “BnC“ is an abbreviation for “ball-and-chain” motif, which is an important N-terminal regulatory motif able to close the entrance to the active site.

Some typos appear in the text: “apo enzyme”, “conduced” etc.

Response: We have re-examined the whole manuscript for typos and additional words (see response to other reviewers)

Why is citation 14 in italic?

Response: There is no specific reason and the incorrect format has been changed.

Conclusions of this manuscript are in accordance with the data presented. The significance of this manuscript i.e., the elucidation of interaction of DHS with its substrate eIF5A will support the discovery of small molecules against the human DHS protein. Hitherto, only spermidine mimetics have been applied which are not target specific.

Reviewer #4 (Remarks to the Author):

In this manuscript, Wator and colleagues aimed to discover the molecular details of the deoxyhypusination process and to provide insights into the effects of deoxyhypusine synthase (DHS) mutations. To this end, they elucidated the cryo-EM structure of eIF5A in complex with DHS. The model illustrates how eIF5A, specifically the hypusine-containing loop, is recognized at the active site. The authors later used revealed interactions to generate DHS mutants and characterized them for stability and activity of DHS. Furthermore, they investigated clinically-relevant DHS mutations and disclosed their impact on eIF5A interaction. Remarkably, the authors also captured DHS in a reaction intermediate and presented its X-ray crystal structure. Using both structures they proposed how deoxyhypusination reaction takes place at the molecular level. The manuscript is exciting and provides valuable insights into the deoxyhypusination process, one of the ancient and essential reactions. The experiments use state of the art techniques. The data is verified by complementary methods and, in general, supports the conclusion drawn. Several points should be addressed before publication.

1) The manuscript will greatly benefit from having a close-up view of eIF5A (and hypusine/ lysine loop) and the surrounding active site interactions. Figures 1c and 2a (even Fig. 2c) have binding site views but each presents side chains of a single component. The supplementary video beautifully depicts such a recognition. However, given that the eIF5A recognition is one of the most important results of the study, presenting such an interface directly as a main display item will add strength to the manuscript.

Response: We are grateful for the suggestions, which prompted us to further improve our figures. We followed the suggestions and we are very happy with the result. We updated Fig. 2a, which now contains two panels that improve the structural interpretation of our results. In the left panel, side chains of the eIF5A hypusine loop are presented as sticks and the DHS is presented as a

transparent cartoon with the colored position of the residues interacting with the hypusine loop. In the right panel, the side chains of the DHS interface are presented as sticks and eIF5A is presented as a transparent cartoon.

2) Accordingly, Figure 2a should be moved to Figure 3 with a similar orientation as in Fig. 3f-h. Otherwise, readers will need to scroll back and forth to locate where the mentioned mutations (mentioned in Fig.3) are.

Response: As mentioned above we have replaced Fig. 2a with an updated version. Furthermore, we have divided it into two separate figures which enhance the interpretability (see response above). However, we would prefer refraining from creating the new figure panel depicting all the mentioned residues (or moving Fig 2a to new Fig 4) as in our opinion it would create unnecessary redundancy. Of note, our atomic models and maps will be freely available via the PDB and EMDB after acceptance and the scientific community will be able to analyze the specific conformation of each residue individually.

3) The cryo-EM structure shows a single eIF5A bound to tetrameric DHS, which looks in line with previous findings. Given the similarities and differences to DHS apo structures, is there any structural explanation why a single eIF5A (out of 4 binding sites) is present? Could this be influenced by the sample preparation? For example, was there a stoichiometric or excess eIF5A during complex formation? In any case, sample preparation should be described in the methods section.

Response: We apologize for not including a detailed description of the complex reconstitution in the initial version of the manuscript (please see the detailed response to reviewer 1). We have added the following section in the Method parts of the revised manuscript, accordingly

Complex reconstitution

We reconstituted the human eIF5A-DHS^{K329A} complex using previously purified individual components, namely eIF5A-1 and catalytically inactive DHS^{K329A}. DHS^{K329A} was mixed with a molar excess of eIF5A-1 (2 mole eIF5A per 1 mole DHS). Proteins were incubated in 200 mM Glycine/NaOH pH=9.3 buffer with 200 mM NaCl in the presence of 1 mM NAD and 1 mM SPD. The complex was then purified by size-exclusion chromatogram using a Superdex200 Increase 10/300 column (GE Healthcare) pre-equilibrated with 200 mM Glycine/NaOH pH=9.3 buffer with 200 mM NaCl. Due to the very small difference in mass of the complex relative to DHS alone the fraction from the very front of the elution peak was used for further investigation.

Furthermore, referring to the raised issues of the stoichiometry of the complex, our analyses, as rightly noted by the reviewer, are in agreement with previously reported data. We have very carefully analyzed the structural differences between the cryo-EM structures of the complex and the crystal structures of the apo DHS. The largest, noticeable difference is that three *ball and chain* motifs closing access to the ligand-free active centers of the DHS are visible in the eIF5A-DHS^{K329A} structure. In the previously described crystal structures of DHS, the *ball and chain* motif was either invisible, presumably due to its mobility/flexibility, or only two were visible, while the other two active site entrances have been exposed (open). Thus, it appears that the binding of eIF5A through subtle conformational changes in the DHS structure (the differences in angles measured between DHS protomers that we described in the manuscript) causes *the ball and chain* motif to close the vacant active site entrances.

One wonders whether the fact that the complex exists in a stoichiometry of 4 molecules (1 tetramer) of DHS to 1 molecule of eIF5A is due to the way the sample is prepared. In our studies, we used different molar ratios of the components of the complex and different protein variants,

both wtDHS and DHS^{K329A}. Regardless of the molar ratios of DHS to eIF5A used, our chromatographic analyses did not differ much from each other (even when using a 4-fold excess of eIF5A) and we surprisingly could not achieve saturation of DHS by eIF5A and non-complexed DHS was always present. Nevertheless, it is worth mentioning that using wild-type DHS we were not able to obtain a satisfactory cryo-EM structure and our micrographs only showed DHS molecules without bound/visible eIF5A at a rather low resolution ($>5\text{\AA}$). Our explanation for this is that the complex formed using wild-type protein variants is dynamic and there could be a fairly rapid exchange and dissociation of eIF5A. Only when we used the catalytically inactive DHS variant did the dynamics of the complex formation apparently stabilize.

Furthermore, it should be mentioned that a major limitation of our study is that it was carried out *in vitro* using purified components of the more complex system. In the scenario of the deoxyhypusination reaction taking place in the living cell, not only do we have to deal with additional components that may influence the formation of the complex (e.g. interactions with DOHH, the ribosome or the kinases CK2, ERK1/2, etc.). Some studies have also hypothesized that in a DOHH-deficient scenario, DHS is constantly linked to eIF5A (in the cell) (Pallmann et al JBC 2015). It seems that studies using the cellular system and, for example, high-resolution microscopy like MINFLUX could help to elucidate the exact mechanism and stoichiometry of the complex in the cellular system and this is certainly an issue we will want to address experimentally in the future.

4) Figure 3d and 3e need revision. At the moment, color coding and labelling in legends are inconsistent. Similarly, coloring of the curves look mixed up. For instance, in Fig. 3d, there is no green curve corresponding to D243A. The authors should make sure similar errors are not present in 3e. In addition, color choice for the curves is not ideal. Similar shades were often used for curves that are far apart. The authors should reconsider their palette choice or simplify the coloring. Alternatively, curves from left to right could be in the same order as legends top to bottom or so.

Response: We agree that in the previous version of the manuscript the graphs were difficult to follow. We have now made new graphs to replace previous panels d and e. We believe that the current selection of colors facilitates the distinction between the individual measurements.

5) In HDX-MS experiments, changes in deuterium exchange rates could hint conformational alterations or they could simply indicate increased or decreased stability upon binding of the interacting residues. How can one separate which one is the predominant one? How can the authors be so sure that this leads to “folding/unfolding of secondary structures” (Line 140-141) in the context of eIF5A-DHS interaction. They suggest for example that C-terminal of eIF5A undergoes conformational changes upon binding to DHS. This region does not interact with DHS, so can they comment on how and why such a folding/unfolding occurs? In general, findings of this section will benefit from further discussion.

Response: We agree, that the previous results section on the HDX-MS data was not adequate to describe all observations from the experiment. We amended the text and now highlight in which areas HDX differences appear and which of these are also linked to the appearance of mixed populations/bimodal behavior of HDX. The appearance of bimodal behavior of deuterium incorporation of peptides in literature is often linked to the transition between different conformations (given that e.g., no protein degradation obscures the data), which we intended to state by “folding/unfolding of secondary structures”. As for the changes in HDX in the C-terminal portion of eIF5A, we can at this point not provide a plausible explanation for the increased HDX in the context of the complex as this part was poorly resolved in our cryo-EM structure of eIF5A-DHS. The degree of bimodality in this eIF5A C-terminal entity is very similar between both states

so we believe that transitions between structural ensembles occur both in individual eIF5A as well as DHS-bound eIF5A. As for DHS, we believe that the bimodal behavior may, at least partially, arise from the 1:4 stoichiometry of eIF5A-DHS, which should leave some DHS molecules in the sample ‘untouched’ by eIF5A. Unfortunately, HDX-MS cannot discriminate between eIF5A-bound DHS molecules and non-bound ones.

6) In Extended Data Fig. 5e and f, the font size is too small to read. Accordingly, it is hard to distinguish the colors and corresponding structure segments in panels a, b and d.

Response: We increased the font sizes in all figures related to HDX-MS experiments.

7) Extended Data Fig. 7 should be renumbered as Data Fig. 6 to keep the logical order in the text.

Response: After the revision, the numbering of Figures was updated and checked.

8) In line 190-192, “Mutations in active site residues (H288A and K329A), had no or only minor effects on eIF5A binding (K_D 49.7 ± 28.7 nM and 128.6 ± 37.9 nM)”, reduction of K329A mutant’s affinity to 128 nM is considered as “only minor”. While in line 316-318, similar reduction in N173S and deletion mutants were considered significant and led to the further conclusions. One of these statements require revision.

Response: We apologize for this inconsistency. We amended the said paragraph, which now reads as follows –

The mutation in active site residue H288A did not affect the eIF5A binding (K_D 49.7 ± 28.7 nM). On the other hand, the K329A substitution showed a decrease in affinity (K_D 128.6 ± 37.9 nM; Fig. 3d, Extended Data Fig. 8). Moreover, mutation of the Trp327 residue caused a significant decrease in affinity to 602.5 ± 284.2 nM, highlighting the importance of this residue for complex formation.

9) Quality of crystallographic data look very nice. Nevertheless, RSRZ outliers seem to be an issue in all three X-ray structures. So, I wonder if these point to a general problem in the refinement strategy. Details of the validation reports reflect significant number of outliers in fit of the model and data, and even include some of the ligands. I don’t believe that they will affect the conclusions of the text, yet I invite authors to investigate potential problems either in the data or in the refinement approach.

Response: The real-space R-value (RSR) is a measure of the quality of fit between a part of an atomic model (in this case, one residue) and the data in real space. This means, that if there is a weak density around atoms comprising a given residue it will likely be an outlier. This is often the case if the local map indicated significant main chain flexibility, particularly at the solvent-exposed surface and close to „chain breaks” adjacent to flexible loops. Another approach to deal with this would be to delete all residues or their sidechains with weak local electron density at a cost of model completeness and integrity. In the presented structures we retained all sidechains and as many residues as possible for realistic modelling maintaining good geometry and refinement statistics. In our opinion, the absence of side-chain atoms is very confusing to non-structural biologists. Most of the RSRZ outliers are located in the region 77-87 which is highly flexible in all our structures (this study and Wator et al., *Biomolecules* 2020) and the N-terminus. Of note, our atomic models and maps will be freely available via the PDB and EMDB after acceptance.

10) Crystallographic table: all numbers should consistently have 1 or 2 decimal digits.

Response: We have reviewed and corrected our crystallographic table for decimal digit consistency for each parameter. We kept 3 decimal digits in case of rms(bond) values. Otherwise, we followed the IUCr Commission on Crystallographic Nomenclature recommendations.

11) In electrostatic potential images, blue and red generally refer to max and min, respectively, since they are also used to depict nitrogen and oxygen atoms. Using inverse coloring in Figure 1d could mislead readers, although it is explicitly explained in the color key and also in the legends.

Response: We thank the reviewer for this comment. The coloring scheme was applied according to the default setting in ChimeraX software. The blue refers to positive local potential and the red to negative local electrostatic potential. We are sorry for the misleading legend inserted into the first version of the panel. We have now corrected the colors (Fig. 1d) and the description in the figure legend.

Typos/Grammar

12) Line 97 Likewise (no However)

13) Line 148 homo-tetramer (no underscore)

Response: Both issues have been resolved in the revised version

14) **Supplementary Line 47 (a, c) Differences in HDX of eIF5A (not a and b)**
Line 49 (b, d) areas of eIF5A... (not c and d)

Response: The supplementary figures have been highly amended, accordingly.

REVIEWERS' COMMENTS

Reviewer #1 (Remarks to the Author):

I thank the authors for satisfactorily answering my queries and adjusting the manuscript accordingly.

Reviewer #2 (Remarks to the Author):

After reading the revised manuscript "Cryo-EM structure of human eIF5A-DHS complex reveals the molecular basis of hypusination-associated neurodegenerative disorders" by Dr Grudnik and colleagues together with reviewers' responses the reviewer is satisfied with the replies and changes made to the manuscript as they significantly increased the value of the presented data. The reviewer now suggests the paper for acceptance to publish in Nat. Commun.

Reviewer #3 (Remarks to the Author):

The manuscript by Wator et al. describes the molecular mechanism how eIF5A interacts with DHS in the first step of hypusine modification. The authors applied cryo EM to crystallize the complex. The obtained data are excellent and a continuation of Dr. Park's DHS crystallization experiments. One important finding is that one eIF5A monomer interacts with the DHS substrate by unlocking the ball and chain motif at the active site of the enzyme. Moreover, studies with a set of mutations defined essential amino acids for complex formation likewise the Trp327 residue. Interestingly, K329A in the active site of DHS does not affect the interaction with eIF5A. In the second part of the manuscript the authors studied the reaction mechanism in two disease associated variants causing neurological disorders. The deletion variant exhibited a misfolding of the protein while the DHS NS173 variant had a less pronounced impact on stability and activity. The work has a strong impact on the polyamine field and a translational aspect with respect to therapy.

The revised version of the manuscript has strongly improved and the authors have clarified and corrected all the the previous concerns. No additional evidence is needed.

The material and methods part provides details, legends have been corrected and are more self explaining.

The conclusion and interpretation of data is justified.

The MS combines basic research and translational aspects and therefore earns publication.

REVIEWERS' COMMENTS

Reviewer #1 (Remarks to the Author):

I thank the authors for satisfactorily answering my queries and adjusting the manuscript accordingly.

Reviewer #2 (Remarks to the Author):

After reading the revised manuscript "Cryo-EM structure of human eIF5A-DHS complex reveals the molecular basis of hypusination-associated neurodegenerative disorders" by Dr Grudnik and colleagues together with reviewers' responses the reviewer is satisfied with the replies and changes made to the manuscript as they significantly increased the value of the presented data. The reviewer now suggests the paper for acceptance to publish in Nat. Commun.

Reviewer #3 (Remarks to the Author):

The manuscript by Wator et al. describes the molecular mechanism how eIF5A interacts with DHS in the first step of hypusine modification. The authors applied cryo EM to crystallize the complex. The obtained data are excellent and a continuation of Dr. Park's DHS crystallization experiments. One important finding is that one eIF5A monomer interacts with the DHS substrate by unlocking the ball and chain motif at the active site of the enzyme. Moreover, studies with a set of mutations defined essential amino acids for complex formation likewise the Trp327 residue. Interestingly, K329A in the active site of DHS does not affect the interaction with eIF5A. In the second part of the manuscript the authors studied the reaction mechanism in two disease associated variants causing neurological disorders. The deletion variant exhibited a misfolding of the protein while the DHS NS173 variant had a less pronounced impact on stability and activity.

The work has a strong impact on the polyamine field and a translational aspect with respect to therapy.

The revised version of the manuscript has strongly improved and the authors have clarified and corrected all the the previous concerns. No additional evidence is needed.

The material and methods part provides details, legends have been corrected and are more self explaining.

The conclusion and interpretation of data is justified.

The MS combines basic research and translational aspects and therefore earns publication.

Authors' reply

We thank the reviewers for their remarks and comments.